# LINEARITY OF RELATION DECODING IN TRANSFORMER LANGUAGE MODELS

Evan Hernandez[1]*      Arnab Sen Sharma[2]*      Tal Haklay[3]      Kevin Meng[1]
Martin Wattenberg[4]      Jacob Andreas[1]      Yonatan Belinkov[3]      David Bau[2]

## ABSTRACT

Much of the knowledge encoded in transformer language models (LMs) may be expressed in terms of relations: relations between words and their synonyms, entities and their attributes, etc. We show that, for a subset of relations, this computation is well-approximated by a single linear transformation on the subject representation. Linear relation representations may be obtained by constructing a first-order approximation to the LM from a single prompt, and they exist for a variety of factual, commonsense, and linguistic relations. However, we also identify many cases in which LM predictions capture relational knowledge accurately, but this knowledge is not linearly encoded in their representations. Our results thus reveal a simple, interpretable, but heterogeneously deployed knowledge representation strategy in LMs.

## 1    INTRODUCTION

How do neural language models (LMs) represent *relations* between entities? LMs store a wide variety of factual information in their weights, including facts about real world entities (e.g., *John Adams was elected President of the United States in 1796*) and common-sense knowledge about the world (e.g., *doctors work in hospitals*). Much of this knowledge can be represented in terms of relations between entities, properties, or lexical items. For example, the fact that Miles Davis is a trumpet player can be written as a **relation** (*plays the instrument*), connecting a **subject entity** (*Miles Davis*), with an **object entity** (*trumpet*). Categorically similar facts can be expressed in the same type of relation, as in e.g., (*Carol Jantsch*, *plays the instrument*, *tuba*). Prior studies of LMs (Li et al., 2021; Meng et al., 2022; Hernandez et al., 2023) have offered evidence that subject tokens act as keys for retrieving facts: after an input text mentions a subject, LMs construct enriched representations of subjects that encode information about those subjects. Recent studies of interventions (Hase et al., 2023) and attention mechanisms (Geva et al., 2023) suggest that the mechanism for retrieval of specific facts is complex, distributed across multiple layers and attention heads. Past work establishes *where* relational information is located: LMs extract relation and object information from subject representations. But these works have not yet described *what computation* LMs perform while resolving relations.

In this paper, we show that LMs employ a simple system for representing a portion of their relational knowledge: they implicitly implement (an affine version of) a **linear relational embedding** (LRE) scheme (Paccanaro & Hinton, 2001). Given a relation $r$ such as *plays the instrument*, a linear relational embedding is an affine function $LRE(\mathbf{s}) = W_r\mathbf{s} + b_r$ that maps any subject representation $\mathbf{s}$ in the domain of the relation (e.g., *Miles Davis*, *Carol Jantsch*) to the corresponding object representation $\mathbf{o}$ (e.g., *trumpet*, *tuba*). In LMs, the inputs to these implicit LREs are hidden representations of subjects at intermediate layers, and their outputs are hidden representations at late layers that can be decoded to distributions over next tokens. Thus, a portion of transformer LMs' (highly non-linear) computation can be well-approximated linearly in contexts requiring relation prediction.

More specifically, we find that for a variety of relations: (a) transformer LMs decode relational knowledge directly from subject entity representations ($\mathbf{s}$ in Figure 1); (b) for each such relation, the decoding procedure is approximately affine (LRE); and (c) these affine transformations can be computed directly from the LM Jacobian on a prompt expressing the relation (i.e. $\partial\mathbf{o}/\partial\mathbf{s}$). However, this is not the only system that transformer LMs use to encode relational knowledge, and we also identify relations that are reliably predicted in LM outputs, but for which no LRE can be found.

---

[1]Massachusetts Institute of Technology, [2]Northeastern University, [3]Technion IIT, [4]Harvard University.
*Equal contribution. Correspondence to: dez@mit.edu, sensharma.a@northeastern.edu.

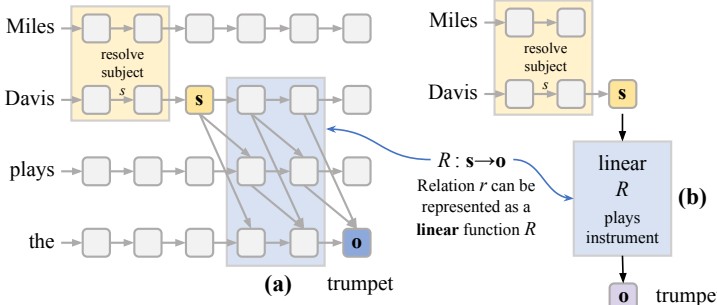

Figure 1: Within a transformer language model, **(a)** how it resolves many relations $r$, such as *plays the instrument*, can be well-approximated by **(b)** a linear function $R$ that maps subject representations **s** to object representations **o** that can be directly decoded.

In GPT and LLaMA models, we search for LREs encoding 47 different relations, covering more than 10k facts relating famous entities (*The Space Needle*, *is located in*, *Seattle*), commonsense knowledge (*banana*, *has color*, *yellow*), and implicit biases (*doctor*, *has gender*, *man*). In $48\%$ of the relations we tested, we find robust LREs that faithfully recover subject–object mappings for a majority of the subjects. Furthermore, we find that LREs can be used to *edit* subject representations (Hernandez et al., 2023) to control LM output.

Finally, we use our dataset and LRE-estimating method to build a visualization tool we call an **attribute lens**. Instead of showing the next token distribution like Logit Lens (nostalgebraist, 2020) the attribute lens shows the *object*-token distribution at each layer for a given relation. This lets us visualize where and when the LM finishes retrieving knowledge about a specific relation, and can reveal the presence of knowledge about attributes even when that knowledge does not reach the output.

Our results highlight two important facts about transformer LMs. First, some of their implicit knowledge is represented in a simple, interpretable, and structured format. Second, this representation system is not universally deployed, and superficially similar facts may be encoded and extracted in very different ways.

## 2 BACKGROUND: RELATIONS AND THEIR REPRESENTATIONS

### 2.1 REPRESENTATIONS OF KNOWLEDGE IN LANGUAGE MODELS

For LMs to generate factually correct statements, factual information must be represented somewhere in their weights. In transformer LMs, past work has suggested that most factual information is encoded in the multi-layer perceptron layers (Geva et al., 2020). These layers act as key–value stores, and work together across multiple layers to enrich the representation of an entity with relevant knowledge (Geva et al., 2022). For instance, in the example from Figure 1, the representation **s** of *Miles Davis* goes through an *enrichment* process where LM populates **s** with the fact that he plays the trumpet as well as other facts, like him being born in Alton, IL. By the halfway point of the LM's computation, **s** contains all the information needed to predict a fact about the subject entity when the LM is prompted to retrieve it.

Once **s** is populated with relevant facts, the LM must decode the fact most relevant to its current prediction task. Formally, a language model is a distribution $p_{\text{LM}}(x)$ over strings $x$, so this information must be retrieved when the LM is prompted to decode a specific fact, such as when it estimates $p_{\text{LM}}(\cdot \mid$ *Miles Davis plays the*). Internally, the object must be decoded and written into the final representation **o** before the next word (*trumpet*) is predicted. Techniques like the logit lens (nostalgebraist, 2020) and linear shortcut approaches (Belrose et al., 2023; Din et al., 2023) reveal that the LM's final prediction can be read off of **o** well before the final layer, and recent work (Geva et al., 2023) suggests that this occurs because specific attention heads (before the final layer) specialize in reading specific relations. Meanwhile, prior work studying the structure of **s** suggests that even though transformers are complex, non-linear neural networks, attributes of entities can be *linearly* decoded from their representations (Li et al., 2021; Hernandez et al., 2023).

But *how transformer LMs themselves* map from enriched entity representations to language-based predictions has remained an open question. Here, we will show that for a subset of relations the transformer LMs implement the learned readout operation in a near-linear fashion.

## 2.2 Neural representations of relations

Why might we expect a linear representation scheme for relational information in the first place? Separate from (and largely prior to) work on neural language models, a long line of artificial intelligence research has studied how to represent relational knowledge. A classic symbolic approach is to encode relational knowledge triplets of the form *(subject, relation, object)*. For example, one might express the fact that Rome is the capital of Italy as *(Rome, is-capital-of, Italy)*. This triplet format is extremely flexible, and has been used for a variety of tasks (Richens, 1956; Minsky, 1974; Lenat, 1995; Miller, 1995; Berners-Lee et al., 2001; Bollacker et al., 2008).

While representing relational triplets symbolically is straightforward, it is far less clear how to embed these relational structures in deep networks or other connectionist systems. Surveys (Ji et al., 2021; Wang et al., 2017) list more than 40 techniques. These variations reflect the tension between the constraints of geometry and the flexibility of the triplet representation. In many approaches, subject and object entities $s$ and $o$ are represented as vectors $\mathbf{s} \in \mathbb{R}^m, \mathbf{o} \in \mathbb{R}^n$; for a given relation $r$, we define a **relation function** $R : \mathbb{R}^m \to \mathbb{R}^n$, with the property that when $(s, r, o)$ holds, we have $\mathbf{o} \approx R(\mathbf{s})$.

One way to implement $R$ is to use linear transformations to represent relations. For instance, in *linear relational embedding* (Paccanaro & Hinton, 2001), the relation function has the form $R(\mathbf{s}) = W_r \mathbf{s}$ where $W_r$ is a matrix depending on relation $r$. A modern example of this encoding can be seen in the positional encodings of many transformers (Vaswani et al., 2017). More generally, we can write $R$ as an *affine* transformation, learning both a linear operator $W_r$ and a translation $b_r$ (Lin et al., 2015; Yang et al., 2021). There are multiple variations on this idea, but the basic relation function is:

$$R(\mathbf{s}) = W_r \mathbf{s} + b_r. \tag{1}$$

## 3 Finding and Validating Linear Relational Embeddings

### 3.1 Finding LREs

Consider a statement such as *Miles Davis plays the trumpet*, which expresses a fact $(s, r, o)$ connecting a subject $s$ to an object $o$ via relation $r$ (see Figure 1). Within the transformer's hidden states, let $\mathbf{s}$ denote the representation[5] of the subject $s$ (*Miles Davis*) at layer $\ell$, and let $\mathbf{o}$ denote the last-layer hidden state that is directly decoded to get the prediction of the object's first token $o$ (*trumpet*). The transformer implements a calculation that obtains $\mathbf{o}$ from $\mathbf{s}$ within a textual context $c$ that evokes the relation $r$, which we can write $\mathbf{o} = F(\mathbf{s}, c)$.

Our main hypothesis is that $F(\mathbf{s}, c)$ can be well-approximated by a linear projection, which can be obtained from a local derivative of $F$. Denote the Jacobian of $F$ as $W = \partial F / \partial \mathbf{s}$. Then a first-order Taylor approximation of $F$ about $\mathbf{s}_0$ is given by:

$$\begin{aligned} F(\mathbf{s}, c) &\approx F(\mathbf{s}_0, c) + W(\mathbf{s} - \mathbf{s}_0) \\ &= W\mathbf{s} + b, \\ \text{where } b &= F(\mathbf{s}_0, c) - W\mathbf{s}_0 \end{aligned} \tag{2}$$

This approximation would only be reasonable if $F$ has near-linear behavior when decoding the relation from any $\mathbf{s}$. In practice, we estimate $W$ and $b$ as the mean Jacobian and bias at $n$ examples $\mathbf{s}_i, c_i$ within the same relation, which gives an unbiased estimate under the assumption that noise in $F$ has zero value and zero Jacobian in expectation (see Appendix B). That is, we define:

$$W = \mathbb{E}_{\mathbf{s}_i, c_i} \left[ \frac{\partial F}{\partial \mathbf{s}} \bigg|_{(\mathbf{s}_i, c_i)} \right] \quad \text{and} \quad b = \mathbb{E}_{\mathbf{s}_i, c_i} \left[ F(\mathbf{s}, c) - \frac{\partial F}{\partial \mathbf{s}} \mathbf{s} \bigg|_{(\mathbf{s}_i, c_i)} \right] \tag{3}$$

This simple formulation has several limitations that arise due to the use of layer normalization (Ba et al., 2016) in the transformer: for example, $\mathbf{s}$ is passed through layer normalization before contributing to the computation of $\mathbf{o}$, and $\mathbf{o}$ is again passed through layer normalization before leading

---

[5]Following insights from Meng et al. (2022) and Geva et al. (2023), we read $\mathbf{s}$ at the last token of the subject.

to token predictions, so in both cases, the transformer does not transmit changes in scale of inputs to changes in scale of outputs. That means that even if Equation 2 is a good estimate of the direction of change of $F$, it may not be an accurate estimate of the magnitude of change.

In practice, we find that the magnitude of change in $F(\mathbf{s}, c)$ is underestimated in our calculated $W$ (see Appendix C for empirical measurements). To remedy this underestimation we make $W$ in Equation (2) *steeper* by multiplying with a scalar constant $\beta$ ($> 1$). So, for a relation $r$ we approximate the transformer calculation $F(\mathbf{s}, c_r)$ as an *affine* transformation LRE on $\mathbf{s}$:

$$F(\mathbf{s}, c_r) \approx \text{LRE}(\mathbf{s}) = \beta\, W_r \mathbf{s} + b_r \tag{4}$$

## 3.2 Evaluating LREs

When a linear relation operator LRE is a good approximation of the transformer's decoding algorithm, it should satisfy two properties:

**Faithfulness.** When applied to new subjects $s$, the output of $\text{LRE}(\mathbf{s})$ should make the same predictions as the transformer. Given the LM's decoder head $D$, we define the transformer prediction $o$ and LRE prediction $\hat{o}$ as:

$$o = \operatorname*{argmax}_t D(F(\mathbf{s}, c))_t \quad \text{and} \quad \hat{o} = \operatorname*{argmax}_t D(\text{LRE}(\mathbf{s}))_t$$

And we define faithfulness as the success rate of $o \overset{?}{=} \hat{o}$, i.e., the frequency with which predictions made by LRE from only $\mathbf{s}$ match next-token predictions made by the full transformer:

$$\operatorname*{argmax}_t D(F(\mathbf{s}, c))_t \overset{?}{=} \operatorname*{argmax}_t D(\text{LRE}(\mathbf{s}))_t \tag{5}$$

**Causality.** If a learned LRE is a good description of the LM's decoding procedure, it should be able to model **causal** influence of the relational embedding on the LM's predictions. If it does, then *inverting* LRE tells us how to perturb $\mathbf{s}$ so that the LM decodes a different object $o'$. Formally, given a new object $\mathbf{o}'$, we use LRE to find an edit direction $\Delta \mathbf{s}$ that satisfies:

$$\text{LRE}(\mathbf{s} + \Delta \mathbf{s}) = \mathbf{o}' \tag{6}$$

With $\beta = 1$, we can edit $\mathbf{s}$ as follows:[6]

$$\tilde{\mathbf{s}} = \mathbf{s} + \Delta \mathbf{s}, \quad \text{where } \Delta \mathbf{s} = W_r^{-1}(\mathbf{o}' - \mathbf{o}) \tag{7}$$

We obtain $\mathbf{o}'$ from a different subject $s'$ that is mapped by $F$ to $o'$ under the relation $r$. $\tilde{\mathbf{s}}$ here is essentially an approximation of $s'$. Figure 2 illustrates this procedure.

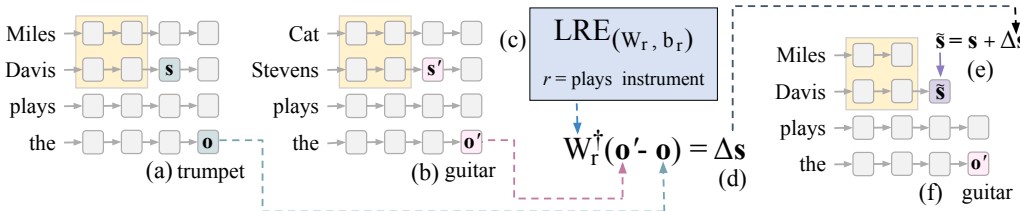

Figure 2: Illustration of the representation editing used to measure *causality*. Under the relation $r = plays$ *the instrument*, and given the subject $s = Miles\ Davis$, LM will predict $o = trumpet$ **(a)**; and given the subject $s' = Cat\ Stevens$, the output is $o' = guitar$ **(b)**. If the computation from $\mathbf{s}$ to $\mathbf{o}$ is well-approximated by LRE parameterized by $W_r$ and $b_r$ **(c)**, then $\Delta \mathbf{s}$ **(d)** should tell us the direction of change from $\mathbf{s}$ to $s'$. Thus, $\tilde{\mathbf{s}} = \mathbf{s} + \Delta \mathbf{s}$ would be an approximation of $s'$ and patching $\tilde{\mathbf{s}}$ in place of $\mathbf{s}$ should change the prediction to $o' = guitar$ **(f)**.

Note that Equation (7) requires inverting $W_r$, but the inverted matrix might be ill-conditioned. To make edits more effective, we instead use a low-rank pseudoinverse $W_r^\dagger$, which prevents the smaller singular values from washing out the contributions of the larger, more meaningful singular values. See Appendix D.2 for details.

---

[6]Full derivation in Appendix D.1

We call the intervention a success if $o'$ is the top prediction of the LM after the edit:

$$o' \stackrel{?}{=} \underset{t}{\operatorname{argmax}} \, D(F(\mathbf{s}, c_r \mid \mathbf{s} := \mathbf{s} + \Delta \mathbf{s})) \tag{8}$$

Note that for both *faithfulness* and *causality* we only consider the *first token of the object* when determining success. Limitations of this approach are discussed in Appendix I.

## 4 EXPERIMENTS

We now empirically evaluate how well LREs, estimated using the approach from Section 3, can approximate relation decoding in LMs for a variety of different relations.

**Models.** In all of our experiments, we study autoregressive language models. Unless stated otherwise, reported results are for GPT-J (Wang & Komatsuzaki, 2021), and we include additional results for GPT-2-XL (Radford et al., 2019) and LLaMA-13B (Touvron et al., 2023) in Appendix H.

**Dataset.** To support our evaluation, we manually curate a dataset of 47 relations spanning four categories: factual associations, commonsense knowledge, implicit biases, and linguistic knowledge. Each relation is associated with a number of example subject–object pairs $(s_i, o_i)$, as well as a prompt template that leads the language model to predict $o$ when $s$ is filled in (e.g., *[s] plays the*). When evaluating each model, we filter the dataset to examples where the language model correctly predicts the object $o$ given the prompt. Table 1 summarizes the dataset and filtering results. Further details on dataset construction are in Appendix A.

**Implementation Details.** We estimate LREs for each relation using the method discussed in Section 3 with $n = 8$. While calculating $W$ and $b$ for an individual example we prepend the remaining $n - 1$ training examples as few-shot examples so that the LM is more likely to generate the answer $o$ given a $s$ under the relation $r$ over other plausible tokens. Then, an LRE is estimated with Equation (3) as the expectation of $W$s and $b$s calculated on $n$ individual examples.

Table 1: Information about the dataset of relations used to evaluate LM relation decoding in LMs. These relations are drawn from a variety of sources. Evaluation is always restricted to the subset of $(s, r, o)$ triples for which the LM successfully decodes $o$ when prompted with $(s, r)$.

| Category | # Rel. | # Examples | # GPT-J Corr. |
|---|---|---|---|
| Factual | 26 | 9696 | 4652 |
| Commonsense | 8 | 374 | 219 |
| Linguistic | 6 | 806 | 507 |
| Bias | 7 | 213 | 96 |

We fix the scalar term $\beta$ (from Equation (4)) once per LM. We also have two hyperparameters specific to each relation $r$; $\ell_r$, the layer after which $\mathbf{s}$ is to be extracted; and $\rho_r$, the rank of the inverse $W^\dagger$ (to check *causality* as in Equation (7)). We select these hyperparameters with grid-search; see Appendix E for details. For each relation, we report average results over 24 trials with distinct sets of $n$ examples randomly drawn from the dataset. LREs are evaluated according to *faithfulness* and *causality* metrics defined in Equations (5) and (8).

### 4.1 ARE LREs FAITHFUL TO RELATIONS?

We first investigate whether LREs accurately predict the transformer output for different relations, that is, how *faithful* they are. Figure 3 shows faithfulness by relation. Our method achieves over $60\%$ faithfulness for almost half of the relations, indicating that those relations are linearly decodable from the subject representation.

We are also interested in whether relations are linearly decodable from $\mathbf{s}$ by any other method. Figure 4 compares our method (from Section 3) to four other approaches for estimating linear relational functions. We first compare with Logit Lens (nostalgebraist, 2020), where $\mathbf{s}$ is directly decoded with the LM decoder head $D$. This essentially tries to estimate $F(\mathbf{s}, c)$ as an *Identity* transformation on $\mathbf{s}$. Next, we try to approximate $F$ as $\text{TRANSLATION}(\mathbf{s}) = \mathbf{s} + b$, where $b$ is estimated as $\mathbb{E}[\mathbf{o} - \mathbf{s}]$ over $n$ examples. This TRANSLATION baseline, inspired by Merullo et al. (2023) and traditional word embedding arithmetic (Mikolov et al., 2013), approximates $F$ from the intermediate representation of the last token of $s$ until $\mathbf{o}$ is generated (Figure 1). Then we compare with a linear regression model trained with $n$ examples to predict $\mathbf{o}$ from $\mathbf{s}$. Finally, we apply LRE on the subject embedding $\mathbf{e}_s$ before initial layers of the LM get to *enrich* the representation.

Figure 4 shows that our method LRE captures LM behavior most faithfully across all relation types. This effect is not explained by word identity, as evidenced by the low faithfulness of $\text{LRE}(\mathbf{e}_s)$. Also,

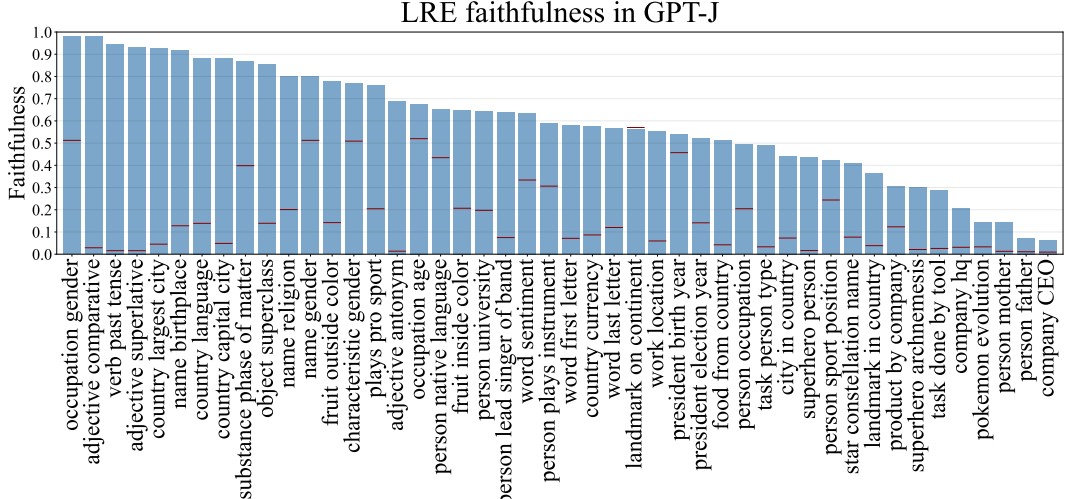

Figure 3: Relation-wise LRE faithfulness to LM computation $F$. Horizontal red lines per relation indicate accuracy of a random-guess baseline. LRE is consistently better than random guess and is predictive of the behavior of the transformer on most relations. However, for some relations such as *company CEO* or *task done by tool*, the transformer LM deviates from LRE, suggesting non-linear model computation for those relations.

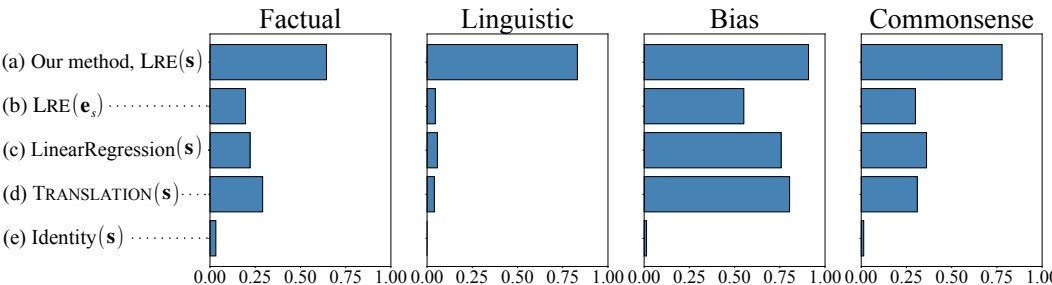

Figure 4: Faithfulness comparison of different linear approximations of LM decoding stratified across different relation types. **(a)** Our method, LRE, applied on **s** extracted after $\ell_r$ **(b)** LRE applied on the subject embedding $\mathbf{e}_s$. The performance different between **(a)** and **(b)** shows the importance of the *enrichment* process **s** goes through in the earlier layers. **(c)** shows the performance of a linear regression model trained with $n = 8$ examples, which is outperformed by a LRE calculated with similar number of examples, $n$. **(d)** is TRANSLATION(**s**) = **s** + $b$, where $b$ is estimated as $\mathbb{E}[\mathbf{o} - \mathbf{s}]$ over $n$ samples. The performance drop in **(d)** compared to **(a)** emphasizes the necessity of the projection term $W$. In **(e)** **s** is directly decoded with the decoder head $D$.

low performance of the TRANSLATION and *Identity* baselines highlight that both the projection and bias terms of Equation (4) are necessary to approximate the decoding procedure as LRE.

However, it is also clear (from Figure 3) that some relations are not linearly decodable from intermediate representations of the subject, despite being accurately predicted by the LM. For example, no method reaches over $6\%$ faithfulness on the *Company CEO* relation, despite GPT-J accurately predicting the CEOs of 69 companies when prompted. This is true across layers (Figure 11 of Appendix E.2) and random sampling of $n$ examples for approximating LRE parameters. This indicates that some more complicated, non-linear decoding approach is employed by the model to make those predictions. Interestingly, the relations that exhibit this behavior the most are those where the range is the names of peoples or companies. One possible explanation is that these ranges are so large that the LM cannot reliably linearly encode them at a single layer, and relies on a more complicated encoding procedures possibly involving multiple layers.

## 4.2 DO LREs CAUSALLY CHARACTERIZE MODEL PREDICTIONS?

We now have evidence that some relations are linearly decodable from LM representations using a first-order approximation of the LM. However, it could be that these encodings are not used by the LM to predict the next word, and instead are correlative rather than causal. To show that LREs causally influence LM predictions, we follow the procedure described in Figure 2 to use the inverse of LRE to change the LM's predicted object for a given subject.

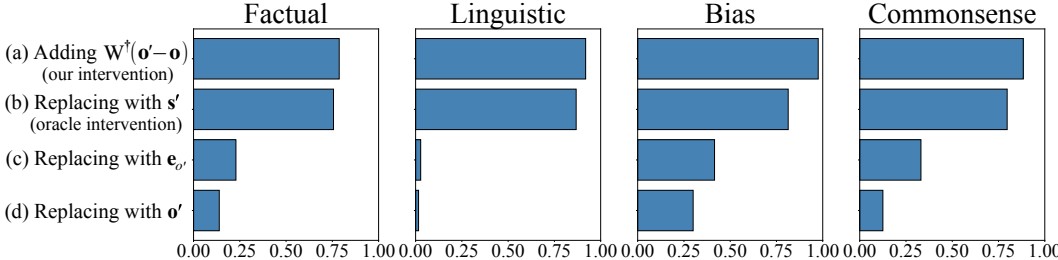

Figure 5: LRE causality compared with different baselines. **(a)** LRE causality on best performing hyperparameters (layer $\ell_r$ and rank $\rho_r$) for each relation $r$. **(b)** is our oracle baseline, inserting the representation $\mathbf{s}'$ of target subject $s'$ in place of $\mathbf{s}$. **(c)** in place of $\mathbf{s}$ inserting $\mathbf{e}_{o'}$, the row in the decoder head matrix $D$ corresponding to $o'$, and **(d)** inserting $\mathbf{o}'$, the output of $F(\mathbf{s}', c_r)$.

In Figure 5 we compare our causality intervention with 3 other approaches of replacing $\mathbf{s}$ such that LM outputs $o'$. If the model computation $F$ from $\mathbf{s}$ to $\mathbf{o}$ is well-approximated by the LRE, then our intervention should be equivalent to inserting $\mathbf{s}'$ in place of $\mathbf{s}$. This direct substitution procedure thus provides an *oracle* upper-bound. Besides the *oracle* approach we include 2 more baselines; in place of $\mathbf{s}$, (1) inserting $\mathbf{o}'$, the output of $F(\mathbf{s}', c_r)$ (2) inserting $\mathbf{e}_{o'}$, the row in the decoder head matrix $D$ corresponding to $o'$ as the embedding of $o'$. These two additional baselines ensure that our approach is not trivially writing the answer $o'$ on the position of $\mathbf{s}$.

Figure 14 of Appendix G.2 compares our method with the baselines for selected relations and across layers. The graphs illustrate how our method matches the *oracle's* performance and differs from the other two baselines. This provides causal evidence that LRE approximates these relations well.

Figure 6 depicts a strong linear correlation between our metrics when the hyperparameters were selected to achieve best causal influence. This means that when an LRE causally influences the LM's predictions, it is also faithful to the model.[7] Also from Figure 6, for almost all relations LRE causality score is higher than its faithfulness score. This suggests that, even in cases where LRE can not fully capture the LM's computation of the relation, the linear approximation remains powerful enough to perform a successful edit.

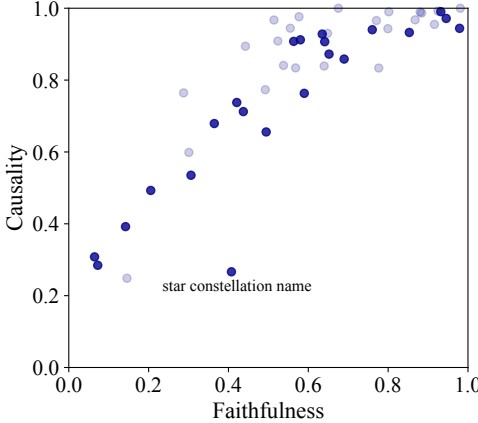

Figure 6: Faithfulness is strongly correlated with causality ($R = 0.84$) when hyperparameters are selected to achieve best causal influence (in GPT-J $\beta = 2.25$). Each dot represents LRE performance for one relation. Bold dots indicate relations for which LRE is evaluated on $\geq 30$ test examples.

While our focus within this work is on the linearity of relation decoding and not on LM representation editing, a qualitative analysis of the post-edit generations reveals that the edits are nontrivial and preserve the LM's fluency; see Table 8 of Appendix G.2.

### 4.3    WHERE IN THE NETWORK DO REPRESENTATIONS EXHIBIT LRES?

In the previous experiments, for each relation we had fixed $\ell_r$ (the layer after which $\mathbf{s}$ is to be extracted) to achieve the best causal influence on the model. However, there are considerable differences in LRE faithfulness when estimated from different layers. Figure 7 highlights an example relation that appears to be linearly decodable from representations in layer 7 until layer 17, at which point faithfulness plummets. Figure 11 of Appendix E.2 shows similar plots for other relations.

Why might this happen? One hypothesis is that a transformer's hidden representations serve a dual purpose: they contain both information about the current word (its synonyms, physical attributes, etc.), and information necessary to predict the next token. At some point, the latter information structure must be preferred to the former in order for the LM to minimize its loss on the next-word prediction task. The steep drop in faithfulness might indicate that a *mode switch* is happening in the

---

[7]However, when the hyperparameters are chosen to achieve best faithfulness we did not notice such strong agreement between faithfulness and causality. Discussion on Appendix E.

LM's representations at later layers, where the LM decisively erases relational embeddings in support of predicting the next word.

Table 2: Example of prompts with and without relation-specific context.

| With relation-specific context |
| --- |
| *LeBron James plays the sport of basketball* |
| *Roger Federer plays the sport of tennis* |
| *Lionel Messi plays the sport of* |

| Without relation-specific context |
| --- |
| *LeBron James basketball* |
| *Roger Federer tennis* |
| *Lionel Messi* |

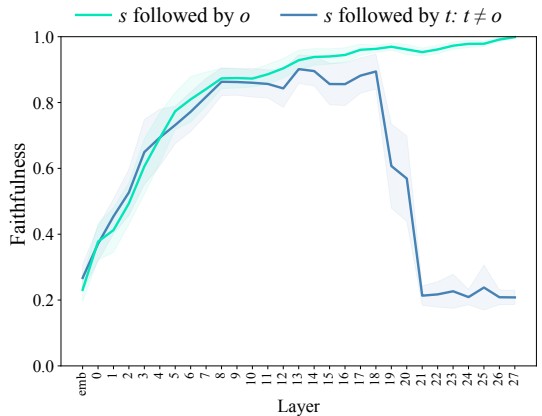

We indeed see in Figure 7 that LRE faithfulness keeps improving in later layers when we remove relation specific texts from our prompt, meaning the $o$ immediately follows $s$ in the prompt (Table 2).

Figure 7: Later layers switching roles from enriching $\mathbf{s}$ to predicting next token. LRE performance across different layers of GPT-J for the factual relation *plays the sport of* with and without relation-specific prompt (Table 2). Faithfulness does not decrease in later layers when $o$ immediately follows the $s$ in the prompt.

## 5 APPLICATION: THE ATTRIBUTE LENS

We apply the LRE to create a novel probing method we call the *attribute lens* that provides a view into a LM's knowledge of an attribute of a subject with respect to a relation $r$. Given a linear function LRE for the relation $r$, the attribute lens visualizes a hidden state $\mathbf{h}$ by applying the LM decoder head $D$ to decode $D(\text{LRE}(\mathbf{h}))$ into language predictions. The attribute lens specializes the Logit Lens (nostalgebraist, 2020) (which visualizes *next token* information in a hidden state $\mathbf{h}$ by decoding $D(\mathbf{h})$) and linear shortcut approaches (Belrose et al., 2023; Din et al., 2023) (where an *affine* probe $A_\ell$ is *trained* to skip computation after layer $\ell$, directly decoding the next token as $D(A_\ell(\mathbf{h}_\ell))$, where $\mathbf{h}_\ell$ is the hidden state after $\ell$). However, unlike these approaches concerned with the immediate next token, the attribute lens is motivated by the observation that each high-dimensional hidden state $\mathbf{h}$ may encode many pieces of information beyond predictions of the immediate next token. Traditional representation probes (Belinkov & Glass, 2019; Belinkov, 2022) also reveal specific facets of a representation, but unlike probing classifiers that divide the representation space into a small number of output classes, the attribute lens decodes a representation into an open-vocabulary distribution of output tokens. Figure 8 illustrates the use of one attribute lens to reveal knowledge representations that contain information about the sport played by a person, and another lens about university affiliation.

This attribute lens can be applied to analyze LM falsehoods: in particular, it can identify cases where an LM outputs a falsehood that contradicts the LM's own internal knowledge about a subject. To quantify the attribute lens's ability to reveal such situations, we tested the attribute lens on a set of 11,891 "repetition distracted" (RD) and the same number of "instruction distracted" (ID) prompts where we deliberately bait the LM to output a wrong $o$, but the LM would have predicted the correct $o$ without the distraction. For example, in order to bait an LM to

Table 3: The performance of the attribute lens on repetition-distracted prompts and instruction-distracted prompts that (almost) never produce the correct statement of a fact. Each row tests 11,891 prompts on GPT-J.

| Condition | R@1 | R@2 | R@3 |
| --- | --- | --- | --- |
| Repetition-distracted prompt | 0.02 | 0.33 | 0.41 |
| Attribute lens on RD prompts | 0.54 | 0.65 | 0.71 |
| Instruction-distracted prompt | 0.03 | 0.17 | 0.25 |
| Attribute lens on ID prompts | 0.63 | 0.73 | 0.78 |

predict that *The capital city of England is... Oslo*, a RD prompt states the falsehood *The capital city of England is Oslo* twice before asking the model to complete a third statement, and an ID prompt states the falsehood followed by the instruction *Repeat exactly.* Although in these cases, the LM will almost never output the true fact (it will predict Oslo instead of London), the attribute lens applied to the last mention of the subject (England) will typically reveal the true fact (e.g., *London*) within the

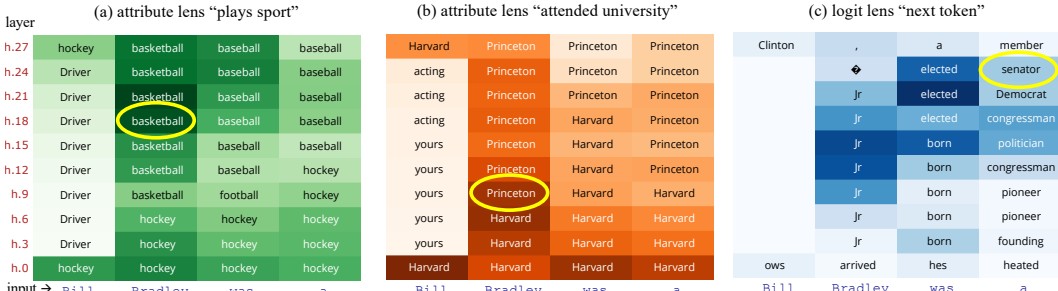

Figure 8: The attribute lens applied to the hidden states of GPT-J processing *Bill Bradley was a*. First two grids visualize the same set of hidden states under the attribute lens for two different relations. The word in each rectangle is the most-likely token in the distribution $D(\text{LRE}(\mathbf{h}))$, where $D$ applies the transformer decoder head; darker boxes correspond to higher probabilities of the top prediction. In (a) the relation is *plays sport*, and in (b) *attended university*, and both these cases reveal high-scoring predictions for attributes on the subject. For comparison, (c) sets LRE $= I$ which produces the Logit Lens (nostalgebraist, 2020) visualization, in which the visualized relation can be thought of as *next token*. (Senator Bill Bradley was formerly a basketball player who went to school at Princeton.)

top 3 predictions. In Table 3 we show performance of the attribute lens to reveal latent knowledge under this adversarial condition.

## 6 Related Work

**Representation probes.** The structure of the information represented within a neural network is a foundational problem that has been studied from several perspectives. One approach is to identify properties encoded representations by training a probing classifier to predict properties from the representations (Ettinger et al., 2016; Shi et al., 2016; Hupkes et al., 2018; Conneau et al., 2018; Belinkov et al., 2017; Belinkov & Glass, 2019). However, such approaches can overestimate the knowledge contained in a network if the classifier learns to solve a task on its own (Belinkov, 2022); the problem can be mitigated by comparing to a control task (Hewitt & Liang, 2019) or by limiting the training of the probe (Voita & Titov, 2020). Our method differs from probing by avoiding the introduction of a training process entirely: we extract the LRE from the LM itself rather than training a new model.

**Knowledge representation.** Ever since emergent neural representations of relations were first observed in the original backpropagation paper (Rumelhart et al., 1986), neural representations of knowledge and relations have been a central problem in artificial intelligence. Section 2 surveys work in this area including knowledge graph embedding (Wang et al., 2017; Yang et al., 2021) and emergent knowledge representations within a transformer language model (Li et al., 2021; Meng et al., 2022; Hase et al., 2023; Hernandez et al., 2023; Geva et al., 2023). This paper builds on past work in by showing that relational aspects of this knowledge are encoded linearly.

**Knowledge extraction.** The most direct way to characterize knowledge in LMs is to prompt or query them directly (Petroni et al., 2019; Roberts et al., 2020; Jiang et al., 2020; Shin et al., 2020; Cohen et al., 2023). However, recent work has suggested that model knowledge and knowledge retrieval may be localized within small parts of a language model (Geva et al., 2020; Dai et al., 2021; Meng et al., 2022; Geva et al., 2023). In this paper we further investigate the localized retrieval of knowledge and ask whether knowledge about relations and objects can be separated, and whether relations are represented as a linear relational embedding.

## 7 Conclusion

Reverse-engineering the full mechanism of an LLM is a daunting task. In this work, we have found that a certain kind of computation, relation decoding, can often be well-approximated by linear relational embeddings. We have also found that some relations are better-approximated as LREs than others; relations that have an easier or harder random baseline fall on either end of the spectrum. We have shown that LREs estimated from a small set of examples lead to faithful representations that are causally linked to the LM's behavior. Furthermore, LRE can be used to provide specialized *attribute lens* on the LM's intermediate computation, even revealing cases of LM falsehoods.

## ETHICS STATEMENT

By revealing and decoding internal model relations before they are explicitly expressed in model output, LREs can potentially be used to provide information about internal biases or errors, and the causal effects could provide a way to mitigate undesired biases. However, such representation-level representation might be only superficial without correcting internal biases in the model; exploring such applications is a natural step for future work.

## REPRODUCIBILITY STATEMENT

The code and dataset are available at `lre.baulab.info`. We include full details about dataset curation in Appendix A. In addition to the experiment details at the beginnings of Sections 4 and 5, we describe hyperparameter sweeps in Appendix E. We ran all experiments on workstations with 80GB NVIDIA A100 GPUs or 48GB A6000 GPUs using HuggingFace Transformers (Wolf et al., 2019) implemented in PyTorch (Paszke et al., 2019).

## ACKNOWLEDGEMENTS

This research has been supported by an AI Alignment grant from Open Philanthropy, the Israel Science Foundation (grant No. 448/20), and an Azrieli Foundation Early Career Faculty Fellowship. We are also grateful to the Center for AI Safety (CAIS) for sharing their compute resources, which supported many of our experiments.

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

# A  RELATIONS DATASET

The dataset consists of 47 relations stratified across 4 groups; *factual*, *linguistic*, *bias*, and *commonsense*. Six of the *factual* relations were scraped from *Wikidata* while the rest were drawn from the COUNTERFACT dataset by Meng et al. (2022). *linguistic*, *bias*, and *commonsense* relations were newly curated by the authors. See Table 4 for details.

Table 4: Number of examples per relation and the count of accurate predictions by different LMs. Each of the examples were tested using $n = 8$ ICL examples ($n = 5$ for LLaMA-13B). Results presented as mean ($\pm$ std) of the counts across 24 trials with different set of ICL examples. For cases where the count of examples accurately predicted by the LM is less then $n$, it was replaced with "—". LRE estimation was not calculated for such cases. **We also do not calculate LRE for LLaMA-13B where $o$ is a year (*president birth year* and *president election year*) as LLaMA tokenizer splits years by digits (see Table 9).

| Category | Relation | # | # Correct | | |
| --- | --- | --- | --- | --- | --- |
| | | | **GPT-J** | **GPT2-xl** | **LLaMA-13B** |
| factual | person mother | 994 | $182.8 \pm 5.8$ | $83.5 \pm 12.3$ | $613.5 \pm 1.5$ |
| | person father | 991 | $206.1 \pm 7.6$ | $109.2 \pm 6.1$ | $675.5 \pm 5.5$ |
| | person sport position | 952 | $243.3 \pm 94.1$ | $199.9 \pm 67.9$ | $200.0 \pm 0.0$ |
| | landmark on continent | 947 | $797.5 \pm 74.7$ | $421.9 \pm 85.3$ | $200.0 \pm 0.0$ |
| | person native language | 919 | $720.2 \pm 31.3$ | $697.3 \pm 16.1$ | $200.0 \pm 0.0$ |
| | landmark in country | 836 | $565.2 \pm 14.2$ | $274.1 \pm 58.2$ | $200.0 \pm 0.0$ |
| | person occupation | 821 | $131.5 \pm 28.5$ | $42.0 \pm 10.3$ | $404.0 \pm 25.5$ |
| | company hq | 674 | $316.0 \pm 10.4$ | $148.8 \pm 36.6$ | $200.0 \pm 0.0$ |
| | product by company | 522 | $366.4 \pm 8.1$ | $262.4 \pm 23.8$ | $421.3 \pm 6.2$ |
| | person plays instrument | 513 | $237.6 \pm 30.7$ | $144.2 \pm 27.6$ | $249.0 \pm 67.7$ |
| | star constellation name | 362 | $276.3 \pm 3.4$ | $210.3 \pm 16.7$ | $200.0 \pm 0.0$ |
| | plays pro sport | 318 | $244.2 \pm 10.8$ | $195.1 \pm 13.1$ | $294.8 \pm 1.2$ |
| | company CEO | 298 | $90.2 \pm 7.2$ | $15.4 \pm 6.1$ | $155.5 \pm 4.7$ |
| | superhero person | 100 | $49.8 \pm 2.0$ | $36.6 \pm 2.4$ | $71.9 \pm 2.3$ |
| | superhero archnemesis | 96 | $21.5 \pm 2.6$ | $12.7 \pm 2.2$ | $40.8 \pm 2.7$ |
| | person university | 91 | $41.0 \pm 1.6$ | $35.4 \pm 1.9$ | $44.8 \pm 2.2$ |
| | pokemon evolution | 44 | $28.7 \pm 3.9$ | $30.3 \pm 1.1$ | $35.7 \pm 0.5$ |
| | country currency | 30 | $19.4 \pm 0.9$ | $20.2 \pm 0.8$ | $22.0 \pm 0.0$ |
| | food from country | 30 | $16.4 \pm 1.1$ | $11.2 \pm 1.4$ | $18.8 \pm 1.4$ |
| | city in country | 27 | $18.0 \pm 0.9$ | $18.0 \pm 1.1$ | $18.1 \pm 0.7$ |
| | country capital city | 24 | $16.0 \pm 0.0$ | $15.5 \pm 0.6$ | $15.3 \pm 0.5$ |
| | country language | 24 | $15.4 \pm 0.6$ | $14.9 \pm 0.6$ | $15.8 \pm 0.4$ |
| | country largest city | 24 | $15.5 \pm 0.5$ | $14.0 \pm 0.8$ | $15.3 \pm 0.5$ |
| | person lead singer of band | 21 | $13.0 \pm 0.2$ | $10.1 \pm 0.9$ | $13.0 \pm 0.0$ |
| | president birth year | 19 | $11.0 \pm 0.0$ | — | ** |
| | president election year | 19 | $9.5 \pm 0.5$ | $9.9 \pm 0.6$ | ** |
| commonsense | object superclass | 76 | $52.7 \pm 1.5$ | $51.5 \pm 2.2$ | $54.4 \pm 1.7$ |
| | word sentiment | 60 | $47.2 \pm 4.1$ | $42.8 \pm 5.1$ | $50.5 \pm 2.0$ |
| | task done by tool | 52 | $30.6 \pm 1.5$ | $25.8 \pm 1.9$ | $33.7 \pm 1.8$ |
| | substance phase of matter | 50 | $30.4 \pm 6.8$ | $34.2 \pm 3.3$ | $40.5 \pm 1.9$ |
| | work location | 38 | $18.2 \pm 2.6$ | $19.7 \pm 2.6$ | $24.7 \pm 2.3$ |
| | fruit inside color | 36 | $10.2 \pm 1.0$ | $9.0 \pm 0.0$ | $17.3 \pm 1.6$ |
| | task person type | 32 | $18.0 \pm 1.2$ | $16.1 \pm 1.5$ | $19.2 \pm 2.1$ |
| | fruit outside color | 30 | $11.7 \pm 2.1$ | $9.6 \pm 0.7$ | $14.6 \pm 1.4$ |
| linguistic | word first letter | 241 | $223.9 \pm 4.5$ | $199.1 \pm 9.8$ | $233.0 \pm 0.0$ |
| | word last letter | 241 | $28.2 \pm 8.2$ | $21.2 \pm 5.3$ | $188.3 \pm 6.7$ |
| | adjective antonym | 100 | $64.0 \pm 2.3$ | $57.5 \pm 2.4$ | $68.5 \pm 1.5$ |
| | adjective superlative | 80 | $70.5 \pm 0.9$ | $64.4 \pm 3.3$ | $70.5 \pm 0.7$ |
| | verb past tense | 76 | $61.0 \pm 4.6$ | $54.0 \pm 3.0$ | $65.8 \pm 3.9$ |
| | adjective comparative | 68 | $59.5 \pm 0.6$ | $57.6 \pm 0.9$ | $60.0 \pm 0.2$ |
| bias | occupation age | 45 | $25.7 \pm 2.6$ | $22.9 \pm 3.8$ | $32.8 \pm 2.6$ |
| | univ degree gender | 38 | — | $21.5 \pm 2.4$ | $24.2 \pm 2.4$ |
| | name birthplace | 31 | $17.1 \pm 2.6$ | $18.0 \pm 1.4$ | $21.4 \pm 1.1$ |
| | name religion | 31 | $17.0 \pm 2.3$ | $15.1 \pm 2.2$ | $19.8 \pm 1.5$ |
| | characteristic gender | 30 | $15.9 \pm 2.7$ | $15.8 \pm 2.2$ | $19.7 \pm 1.2$ |
| | name gender | 19 | $11.0 \pm 0.0$ | $10.7 \pm 0.6$ | $10.8 \pm 0.4$ |
| | occupation gender | 19 | $9.6 \pm 0.8$ | $9.8 \pm 0.7$ | $10.8 \pm 0.4$ |

## B  ASSUMPTIONS UNDERLYING THE LRE APPROXIMATION

Our estimate of the LRE parameters is based on an assumption that the transformer LM implements relation decoding $F(\mathbf{s}, c)$ in a near-linear fashion that deviates from a linear model with a non-linear error term $\varepsilon(\mathbf{s})$ where both $\varepsilon$ and $\varepsilon'$ are zero in expectation over $\mathbf{s}$, i.e., $\mathbb{E}_\mathbf{s}[\varepsilon(\mathbf{s})] = 0$ and $\mathbb{E}_\mathbf{s}[\varepsilon'(\mathbf{s})] = 0$.

$$F(\mathbf{s}, c) = b + W\mathbf{s} + \varepsilon(\mathbf{s}) \tag{9}$$

Then passing to expectations over the distribution of $\mathbf{s}$ we can estimate $b$ and $W$:

$$b = F(\mathbf{s}, c) - W\mathbf{s} - \varepsilon(\mathbf{s}) \tag{10}$$
$$b = \mathbb{E}_\mathbf{s}[F(\mathbf{s}, c) - W\mathbf{s}] - \mathbb{E}_\mathbf{s}[\varepsilon(\mathbf{s})] \tag{11}$$
$$W = F'(\mathbf{s}, c) - \varepsilon'(\mathbf{s}) \tag{12}$$
$$W = \mathbb{E}_\mathbf{s}[F'(\mathbf{s}, c)] - \mathbb{E}_\mathbf{s}[\varepsilon'(\mathbf{s})] \tag{13}$$

Equations 11 and 13 correspond to the bias term $b$ and and projection term $W$ of Equation (3) in the main paper.

## C  IMPROVING THE ESTIMATE OF $F'(\mathbf{s}, c)$ AS $\beta W$

Empirically we have found that $\beta W$ (with $\beta > 1$) yields a more accurate linear model of $F$ than $W$. In this section we measure the behavior of $F'$ in the region between subject representation vectors to provide some evidence on this.

Take two subject representation vectors $\mathbf{s}_1$ and $\mathbf{s}_2$. The projection term of our LRE model $W$, based on the mean Jacobian of $F$ calculated at subjects $\mathbf{s}_i$, yields this estimate of transformer's behavior when traversing from one to the other

$$F(\mathbf{s}_2) - F(\mathbf{s}_1) \approx W(\mathbf{s}_2 - \mathbf{s}_1). \tag{14}$$

We can compare this to an exact calculation: the fundamental theorem of line integrals tells us that integrating the actual Jacobian along the path from $\mathbf{s}_1$ to $\mathbf{s}_2$ yields the actual change:

$$F(\mathbf{s}_2) - F(\mathbf{s}_1) = \int_{\mathbf{s}_1}^{\mathbf{s}_2} F'(\mathbf{s})d\mathbf{s} \tag{15}$$

$$||F(\mathbf{s}_2) - F(\mathbf{s}_1)|| = \int_{\mathbf{s}_1}^{\mathbf{s}_2} \mathbf{u}^T F'(\mathbf{s})d\mathbf{s} \tag{16}$$

Here we reduce it to a one-dimensional problem, defining unit vectors $\mathbf{u} \propto F(\mathbf{s}_2) - F(\mathbf{s}_1)$ and $\mathbf{v} \propto \mathbf{s}_2 - \mathbf{s}_1$ in the row and column space respectively, so that

$$||F(\mathbf{s}_2) - F(\mathbf{s}_1)|| \approx \mathbf{u}^T W \, \mathbf{v}||\mathbf{s}_2 - \mathbf{s}_1|| \tag{17}$$

By taking the ratio between the sides of (17) we can see how the actual rate of change from $\mathbf{s}_1$ to $\mathbf{s}_2$ is underestimated by $W$. Table 5 reports this value for some selected relations.

In practice we find that setting $\beta$ as a constant for an LM (instead of setting it per relation) is enough to attain good performance across a range of relations. Refer to Appendix E for further details.

Table 5: Ratio between the right hand sides of Equation (16) and (17) for some of the relations.

| Relation | Underestimation Ratio |
|---|---|
| plays pro sport | $2.517 \pm 1.043$ |
| country capital city | $4.198 \pm 0.954$ |
| object superclass | $3.058 \pm 0.457$ |
| name birthplace | $4.328 \pm 0.991$ |

## D  CAUSALITY

### D.1  DERIVATION OF EQN 7

Under the same relation $r$ = *person plays instrument*, consider two different $(s, o)$ pairs: *(s = Miles Davis, o = trumpet)* and *(s' = Cat Stevens, o' = guitar)*. If an LRE defined by the projection term $W$ and translation term $b$ is an well-approximation of the model calculation $F(\mathbf{s}, c)$, then

$$\mathbf{o} = \beta\, W(\mathbf{s}) + b \quad \text{and} \quad \mathbf{o}' = \beta\, W(\mathbf{s}') + b$$

Subtracting $\mathbf{o}'$ from $\mathbf{o}$

$$\begin{aligned}
\mathbf{o}' - \mathbf{o} &= \beta W \mathbf{s}' - \beta W \mathbf{s} \\
&= \beta W(\mathbf{s}' - \mathbf{s}) \quad [\text{since } W \text{ is linear}] \\
\Delta \mathbf{s} = \mathbf{s}' - \mathbf{s} &= \frac{1}{\beta}\, W^{-1}(\mathbf{o}' - \mathbf{o})
\end{aligned} \tag{18}$$

We observe that the edit direction $\Delta\mathbf{s}$ needs to be magnified to achieve good edit efficacy. In our experiments we magnify $\Delta\mathbf{s}$ by $\beta$ (or set $\beta = 1.0$ in Equation (18)).

$$\Delta\mathbf{s} = W^{-1}(\mathbf{o}' - \mathbf{o}) \tag{19}$$

## D.2 Why low-rank inverse $W^{\dagger}$ instead of full inverse $W^{-1}$ is necessary?

In practice, we need to take a low rank approximation $W^{\dagger}$ instead of $W^{-1}$ for the edit depicted in Figure 2 to be successful. With $\beta$ set to 1.0,

$$\begin{aligned}
\mathbf{o}' - \mathbf{o} &= W(\mathbf{s}' - \mathbf{s}) \\
\Delta\mathbf{o} &= W\Delta\mathbf{s}
\end{aligned}$$

If we take $U\Sigma V^T$ as the SVD of $W$, then

$$\begin{aligned}
\Delta\mathbf{o} &= U\Sigma V^T \Delta\mathbf{s} \\
U^T \Delta\mathbf{o} &= \Sigma V^T \Delta\mathbf{s}
\end{aligned}$$

Considering $U^T\Delta\mathbf{o}$ as $\mathbf{o}_u$ and $V^T\Delta\mathbf{s}$ as $\mathbf{s}_v$,

$$\mathbf{o}_u = \Sigma\mathbf{s}_v \quad \text{or} \quad \Sigma^{-1}\mathbf{o}_u = \mathbf{s}_v \tag{20}$$

Here, $\Sigma$ maps $\mathbf{s}_v$ to $\mathbf{o}_u$. This $\Sigma$ is a diagonal matrix that contains the non-negative singular values in it's diagonal and zero otherwise. The greater the singular value the more its effect on $\mathbf{s}_v$. However, if we take the full rank inverse of $\Sigma$ while mapping $\mathbf{o}_u$ to $\mathbf{s}_v$ then the inverse becomes dominated by noisy smaller singular values and they wash out the contribution of meaningful singular values. Thus, it is necessary to consider only those singular values greater than a certain threshold $\tau$ or take a low rank inverse $W^{\dagger}$ instead of a full inverse $W^{-1}$.

The significance of different ranks on causality is depicted on Figure 9. We see that the causality increases with a rank up to some point and starts decreasing afterwards. This suggests that, we are ablating important singular values from $\Sigma$ before an optimal rank $\rho_r$ is reached, and start introducing noisy singular values afterwards.

## E Selecting Hyperparameters ($\beta$, $\ell_r$, and $\rho_r$)

We need to select a scalar value $\beta$ per LM since the slope $W$ of the first order approximation underestimates the slope of $F(\mathbf{s}, c)$ (Appendix C). Additionally, we need to specify two hyperparameters per relation $r$; $\ell_r$, the layer after which $\mathbf{s}$ is to be extracted and $\rho_r$, the rank of the low-rank inverse $W^{\dagger}$.

We perform a grid search to select these hyperparameters. For a specific $\beta$, hyperparameters $\ell_r$ and $\rho_r$ are selected to achieve the best causal influence as there is a strong agreement between *faithfulness* and *causality* when the hparams are selected this way. However, when $\ell_r$ and $\rho_r$ are selected to achieve best faithfulness there is a weaker agreement between our evaluation metrics (Figure 10). Appendix E.2 provides an insight on this.

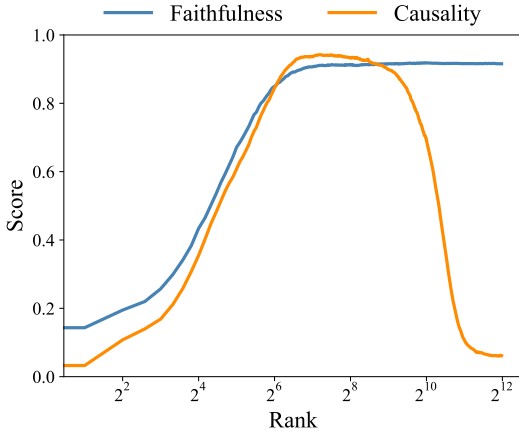
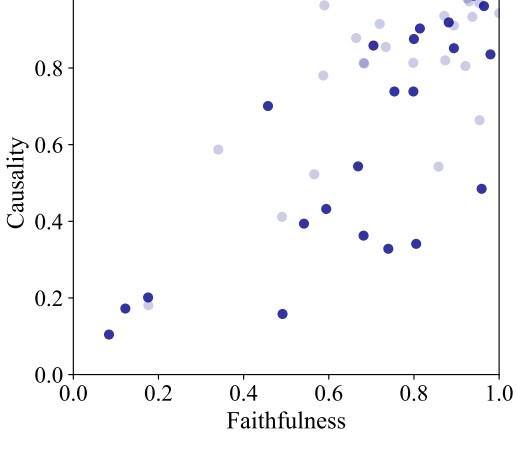

Figure 9: Initially, faithfulness and causality of LRE improve with higher rank. However, after $rank = 2^8$ causality starts declining whereas faithfulness remains stable.

Figure 10: When hparams are selected to achieve best faithfulness there is a weaker correlation of 0.74 between our evaluation metrics unlike in Figure 6 where hparams were selected to achieve best causality. For both this figure and Figure 6 the LM is GPT-J and $\beta = 2.25$.

In sweep over layers we notice that LRE performance per relation increases up to a certain layer and drops afterwards, suggesting a mode-switch in later layers (Figure 11). And, in the sweep over ranks we notice that both edit efficacy and faithfulness increases up to a certain rank. After that edit efficacy starts dropping while faithfulness remains stable. The reasoning behind how causality is affected by higher rank is discussed in Appendix D.2.

## E.1 SELECTING $\beta$

Table 6 represents how performance scores of LRE change with respect to different values of $\beta$ for GPT-J. In our experiments, causality is always calculated with $\beta$ set to 1.0. So, the average causality score remain constant. $\beta$ is selected per LM to achieve the best agreement between our performance metrics *faithfulness* and *causality*. For GPT-J optimal value of $\beta$ is 2.25.

## E.2 LAYER-WISE LRE PERFORMANCE ON SELECTED RELATIONS (GPT-J)

If LRE remains faithful to model up to layer $\ell_{faith}$ it is reasonable to expect that LRE will retain high causality until $\ell_{faith}$ as well. However, an examination of the faithfulness and causality performances across layers reveals that the causality scores drop before $\ell_{faith}$ (Fig. 11). In fact Fig. 14 from Appendix G.2 indicates that all the intervention baselines exhibit a decrease in performance at deeper layers, particularly our method and the *oracle* method at similar layers. This might be a phenomenon associated with this type of intervention in general, rather than a fault with our approximation of the target $\mathbf{s}'$. Notice that, in all of our activation patching experiments we only patch a single state (at the position of the last subject token after a layer $\ell$).

Table 6: Scores achieved by LRE on different values of $\beta$ on GPT-J. $\beta = 2.25$ shows the best correlation between our evaluation metrics. Faithfulness$_\mu$ means the average faithfulness across all the relations (same for Causality$_\mu$).

| $\beta$ | Faithfulness$_\mu$ | Causality$_\mu$ | Corr |
|---|---|---|---|
| 0.00 | 0.17 ± 0.20 | | 0.30 |
| 0.25 | 0.21 ± 0.22 | | 0.33 |
| 0.50 | 0.28 ± 0.24 | | 0.38 |
| 0.75 | 0.36 ± 0.25 | | 0.48 |
| 1.00 | 0.43 ± 0.26 | | 0.58 |
| 1.25 | 0.50 ± 0.26 | | 0.67 |
| 1.50 | 0.54 ± 0.25 | | 0.76 |
| 1.75 | 0.57 ± 0.25 | | 0.80 |
| 2.00 | 0.59 ± 0.25 | | 0.83 |
| **2.25** | 0.59 ± 0.25 | 0.81 ± 0.22 | **0.84** |
| 2.50 | 0.59 ± 0.25 | | 0.84 |
| 2.75 | 0.59 ± 0.25 | | 0.84 |
| 3.00 | 0.59 ± 0.25 | | 0.83 |
| 3.25 | 0.58 ± 0.25 | | 0.81 |
| 3.50 | 0.57 ± 0.25 | | 0.80 |
| 3.75 | 0.56 ± 0.25 | | 0.78 |
| 4.00 | 0.55 ± 0.25 | | 0.76 |
| 4.25 | 0.54 ± 0.25 | | 0.75 |
| 4.50 | 0.53 ± 0.25 | | 0.74 |
| 4.75 | 0.53 ± 0.25 | | 0.72 |
| 5.00 | 0.52 ± 0.25 | | 0.71 |

If the activation after layer $\ell$ is patched, layers till $\ell - 1$ retains information about the original subject $\mathbf{s}$ and they can *leak* information about $\mathbf{s}$ to later layers because of attention mechanism. The deeper

we intervene the more is this *leakage* from previous states and it might reduce the efficacy of these single state activation patching approaches.

It is not reasonable to expect high causality after $\ell_{faith}$, and causality can drop well before $\ell_{faith}$ because of this *leakage*. This also partly explains the disagreement between the two metrics when the hyperparameters are chosen to achieve the best faithfulness (Figure 10).

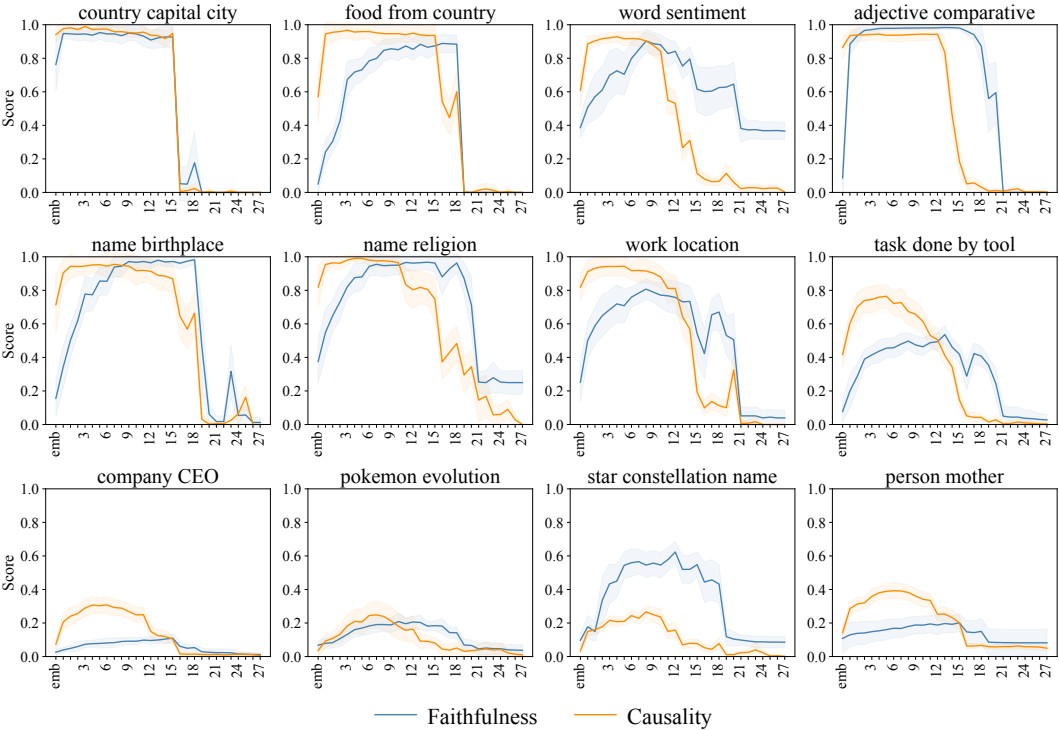

Figure 11: LRE performance for selected relations in different layers of GPT-J. The last row features some of the relations where LRE could not achieve satisfactory performance indicating a non-linear decoding process for them.

## F    VARYING $n$ AND PROMPT TEMPLATE

Figure 12 shows how $lre$ performance changes based on number of examples $n$ used for approximation. For most of the relations both *faithfulness* and *efficacy* scores start plateauing after $n = 5$. In our experiment setup we use $n = 8$ as that is the largest number we could fit for a GPT-J model on a single A6000. However, Figure 12 suggests that a good LRE estimation may be obtained with less number of examples.

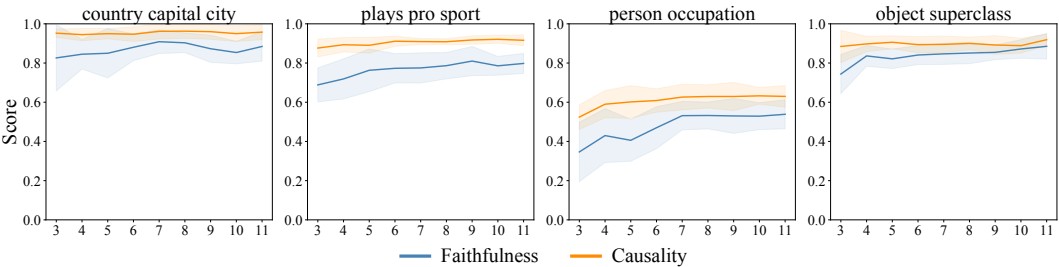

Figure 12: LRE performance across different $n$.

We also test how LRE performance on a relation $r$ changes when the same relation $r$ is contextualized with different prompt templates. Table 7 shows minimal change in *faithfulness* and *causality* scores when LRE is calculated with different prompt templates.

Table 7: LRE performance on different prompt templates. The subject $s$ is inserted in place of {}. Performance scores presented as mean and standard deviation across 24 trials with different sets of training examples.

| Relation | Prompt Template | Faithfulness | Causality |
|---|---|---|---|
| country capital city | The capital of {} is | $0.84 \pm 0.09$ | $0.94 \pm 0.04$ |
| | The capital of {} is the city of | $0.87 \pm 0.08$ | $0.94 \pm 0.04$ |
| | The capital city of {} is | $0.84 \pm 0.08$ | $0.94 \pm 0.04$ |
| | What is the capital of {}? It is the city of | $0.87 \pm 0.07$ | $0.92 \pm 0.05$ |
| plays pro sport | {} plays the sport of | $0.78 \pm 0.07$ | $0.90 \pm 0.03$ |
| | {} plays professionally in the sport of | $0.78 \pm 0.09$ | $0.90 \pm 0.03$ |
| | What sport does {} play? They play | $0.81 \pm 0.06$ | $0.90 \pm 0.03$ |
| person occupation | {} works professionally as a | $0.41 \pm 0.08$ | $0.55 \pm 0.09$ |
| | {} works as a | $0.44 \pm 0.11$ | $0.58 \pm 0.07$ |
| | By profession, {} is a | $0.46 \pm 0.14$ | $0.58 \pm 0.08$ |
| adjective superlative | The superlative form of {} is | $0.93 \pm 0.02$ | $0.97 \pm 0.02$ |
| | What is the superlative form of {}? It is | $0.92 \pm 0.02$ | $0.96 \pm 0.03$ |

## G  BASELINES

### G.1  FAITHFULNESS

In Figure 13 we examine whether our method can be applied to $\mathbf{s}$ extracted from zero-shot prompts that contain only the subject $s$ and no further context. It appears that even when LRE is trained with few-shot examples, it can achieve similar results when applied to $\mathbf{s}$ that is free of any context specifying the relation.

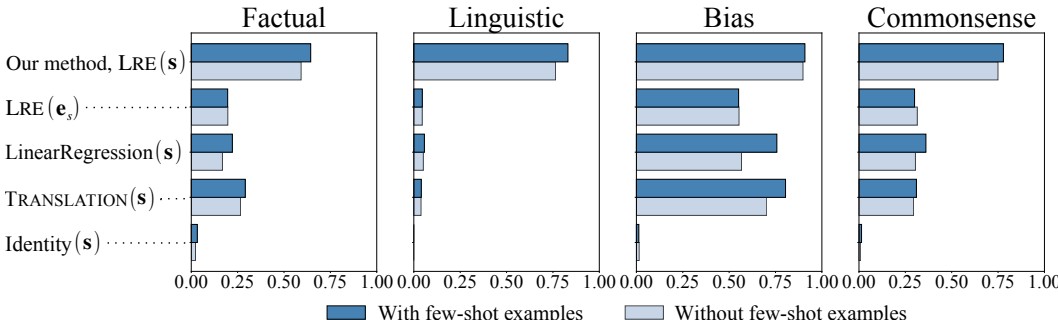

Figure 13: LRE faithfulness on GPT-J compared with different linear functions baselines(same as Figure 4). Each of the functions were approximated with $n = 8$ samples, each prepended with $n - 1$ few-shot examples (Table 2). Dark blue bars indicate faithfulness when evaluated on $\mathbf{s}$ extracted in a similar setup. Light blue bars represent how LRE (trained on few-shot examples) generalize when applied on $\mathbf{s}$ extracted from zero-shot prompts that contain only the subject and no further context.

### G.2  CAUSALITY

Figure 14 shows the LRE causality performance in comparison to other baselines for selected relations and across different layers. If LRE is a good approximation of the model computation, then our causality intervention should be equivalent to the *oracle* baseline, which replaces $\mathbf{s}$ with $\mathbf{s}'$. The graphs demonstrate the similarity between our method performance and oracle performance across all layers. This provides causal evidence that LRE can reliably recover $\mathbf{s}'$ for these relations.

While model editing is not the primary focus of this work, it is still worth examining how our intervention may affect multiple token generation, since it may reveal unexpected side effects of the method. Based on a qualitative analysis of the post-edit generations, it appears that the edits preserve the fluency of the model. Table 8 presents examples of generated texts after intervention.

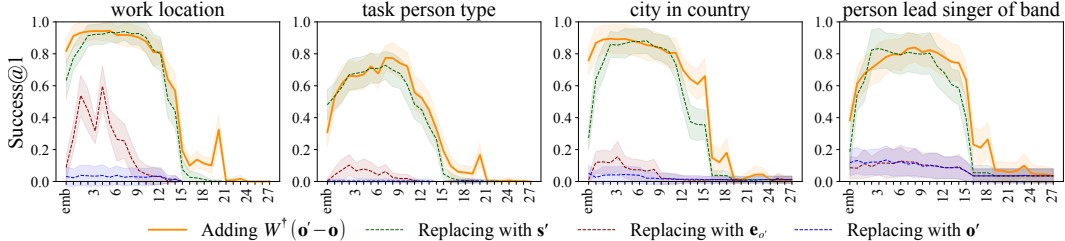

Figure 14: LRE causality across different layers of GPT-J. The causality curve closely matches the peaks and valleys of the *oracle* baseline, replacing **s** with **s**′, suggesting that LRE is a good approximation of the model computation $F$.

Table 8: Generated texts, before and after our *causal* intervention on GPT-J to change its prediction to $o'$.

| Prompt | $o \rightarrow o'$ | Before | After |
|---|---|---|---|
| Miles Davis plays the | trumpet → guitar | trumpet in his band. | guitar live with his band. |
| Siri was created by | Apple → Google | Apple as a personal assistant. | Google and it has become a huge success within Google Maps. |
| Chris Martin is the lead singer of | Coldplay → Foo Fighters | Coldplay and a man of many talents. | Foo Fighters, one of the most successful and popular rock bands in the world. |
| What is the past tense of close? It is | closed → walked | closed. It means it gets closed or is closed. | walked. What is the past tense of read? |

## H LRE ON GPT2-XL AND LLAMA-13B

We provide further results for GPT2-xl and LLaMA-13b to show that autoregressive LMs of different sizes employ this linear encoding scheme for a range of different relations. Figure 15 present LRE performances for each of the three models grouped by relation category.

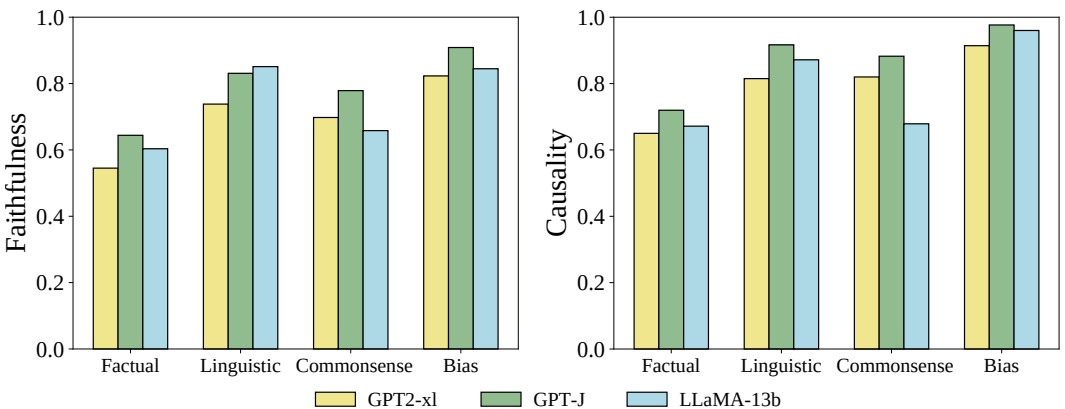

Figure 15: LRE performance in different relation categories on different LMs.

Figures 16a and 16a illustrate the high correlation between our two metrics on GPT2-xl and LLaMA-13b respectively. These findings are consistent with the results reported for GPT-J (Figure 6).

Similarly, Figures 17a and 17b, compare the faithfulness of our method with other approaches of achieving a linear decoding scheme for GPT2-xl and LlaMA.

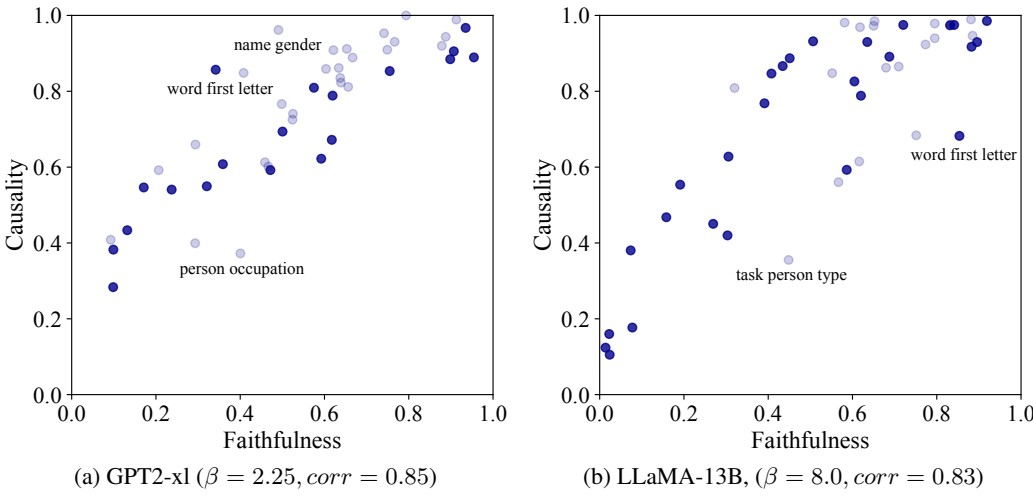

Figure 16: High correlation between faithfulness and causality in both GPT2-xl (R=0.85) and LLaMa-13B (R=0.83). Each of the dots represent LRE performance for a relation. Bold dots indicate relations for which LRE is evaluated on $\geq 30$ test examples.

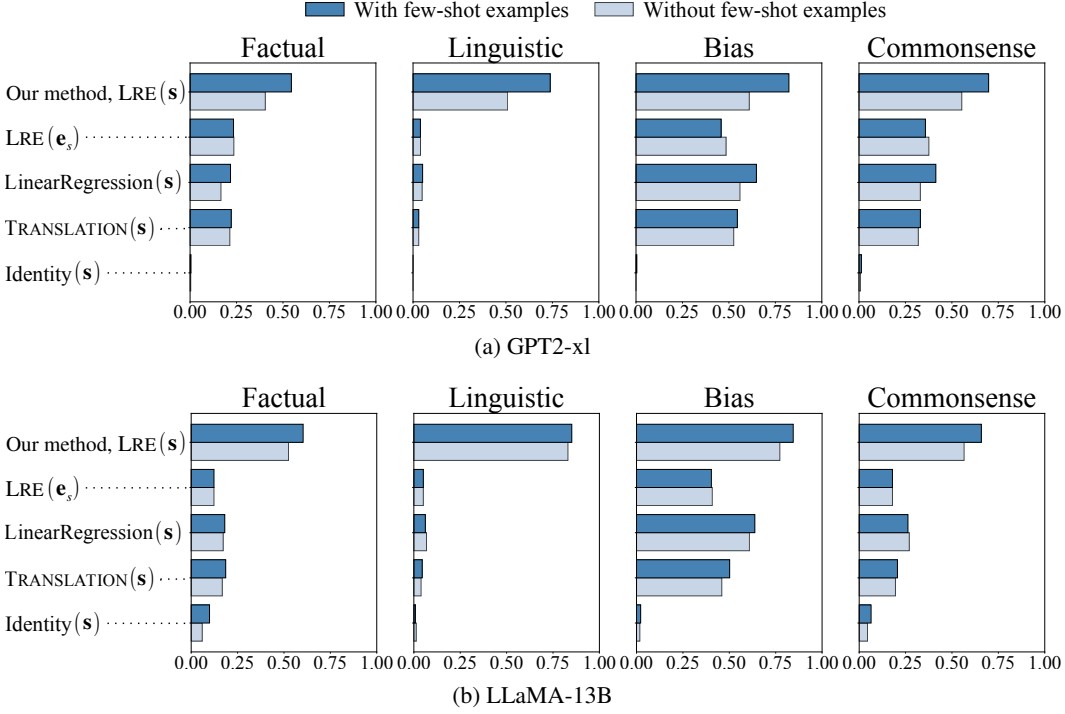

Figure 17: LRE faithfulness on GPT2-xl and LLaMA-13B compared with different baselines. Refer to Figure Figure 4 for details on these baseline approaches of achieving a linear decoding function.

Lastly, Figure 18 depicts LRE faithfulness in GPT2-xl and LLaMA-13b for each of relations in our dataset. According to Spearman's rank-order correlation, GPT-J's relation-wise performance is strongly correlated with both GPT2-xl ($R = 0.85$) and LLaMa-13B ($R = 0.71$), whereas GPT2-xl and LLaMa-13B are moderately corelated ($R = 0.58$).

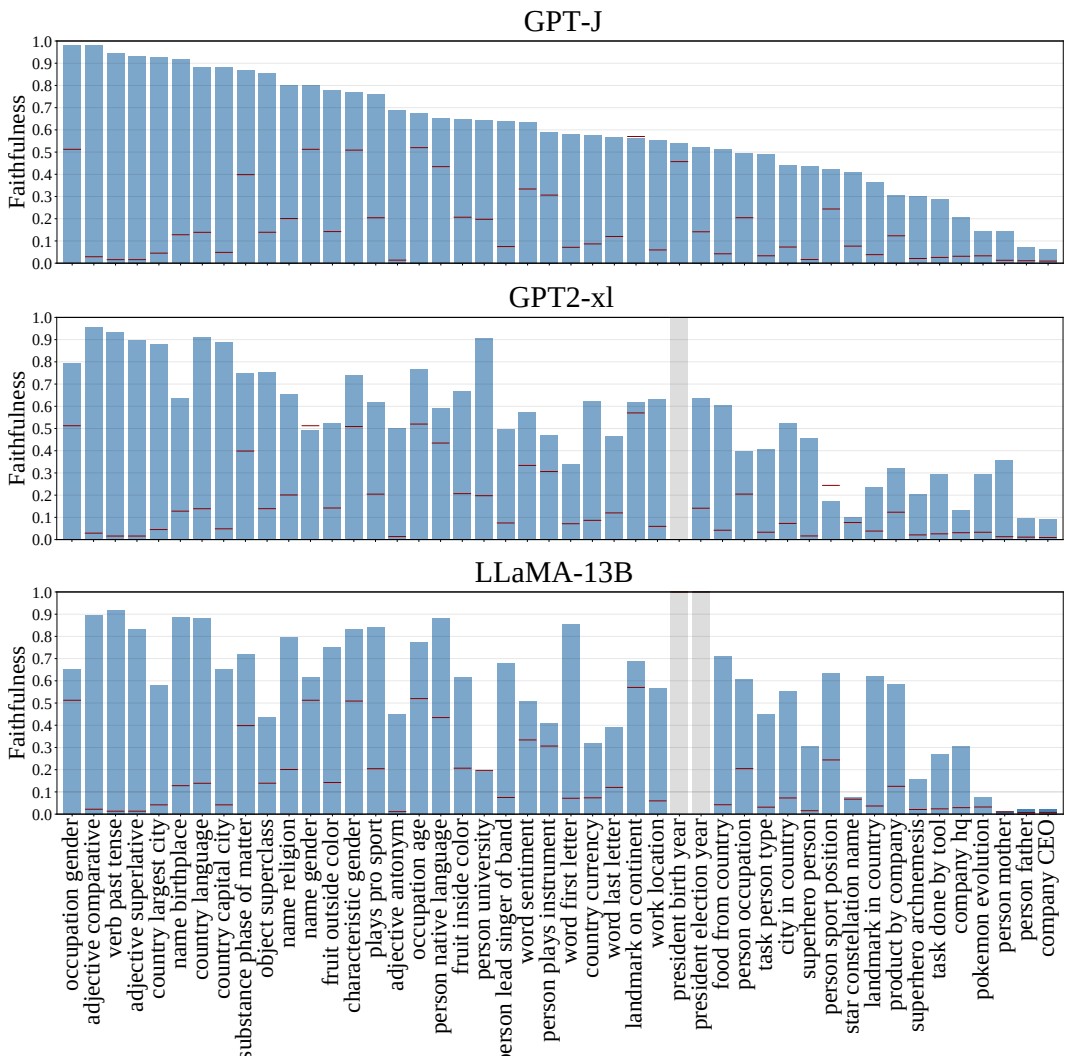

Figure 18: Relation-wise LRE faithfulness to the LM relation decoding function $F$. Horizontal red lines per relation indicate accuracy of a random-guess baseline. Relations are ordered according to their LRE faithfulness in GPT-J. We do not calculate LRE estimation for GPT2-xl on the relation *president birth year* as GPT2-xl can not accurately answer for that relation (Table 4). We also do not calculate LRE of LLaMa-13B for the relations where the $o$ is a year (i.e. *president birth year* and *president election year*) as tokenizer of LLaMA models splits the digits of a year. Such behavior of LLaMA tokenizer makes the relation decoding function trivial, since most of the answers start with "1" for these relations. Those cases were included in this plot (grayed out) to align LRE faithfulness of different LMs by relations.

## I    LIMITATIONS

Our analysis of linear relational embeddings has several core limitations.

**Dataset size**    We have only tested a small set of 47 relations; although we have covered several categories of relations, there are many types of relations we have not explored, such as numerical or physical relations, or logical inferences or multi-hop reasoning.

**First-token correctness criterion**    During all of our experiments, we consider a predicted object correct if the first predicted token matches the first token of the true object. This introduces a risk for false positives: if multiple different objects share a first token, we might erroneously label a prediction as correct. For example, in the relation *person university*, many of the university names start with *"University of"* and such cases would inflate our measurements. We quantify the risk for such false positives in Table 9 and find that many relations have few to no collisions. Nevertheless, the first-token evaluation scheme limits the relations that can be reliably evaluated for linear decoding using this approach.

Table 9: Relation-wise $range$ (count of unique $o$) in our dataset along with the percentage of $o$s uniquely identified by their first token when tokenized with different LM tokenizers. The number inside parenthesis is the actual count of unique first tokens.
$^*$ Tokenizer for LLaMA models splits numeric words by digits. For the relation *president birth year* it only finds one unique first token $\{1\}$, since all the years in this relation start with 1. For *president election year* LLaMA tokenizer finds only two unique first tokens $\{1, 2\}$. Counts for these relations were ignored while calculating average first token coverage for LLaMA-13B,

| Relation | $|range_r|$ | GPT-J | GPT2-xl | LLaMA-13B |
|---|---|---|---|---|
| adjective antonym | 95 | 100.0% (95) | 100.0% (95) | 98.9% (94) |
| adjective comparative | 57 | 100.0% (57) | 100.0% (57) | 93.0% (53) |
| adjective superlative | 79 | 97.5% (77) | 97.5% (77) | 98.7% (78) |
| city in country | 21 | 95.2% (20) | 95.2% (20) | 95.2% (20) |
| company CEO | 287 | 72.5% (208) | 72.5% (208) | 67.6% (194) |
| company hq | 163 | 100.0% (163) | 100.0% (163) | 93.3% (152) |
| country currency | 23 | 100.0% (23) | 100.0% (23) | 91.3% (21) |
| landmark in country | 91 | 100.0% (91) | 100.0% (91) | 97.8% (89) |
| person father | 968 | 41.3% (400) | 41.3% (400) | 38.9% (377) |
| person lead singer of band | 21 | 85.7% (18) | 85.7% (18) | 85.7% (18) |
| person mother | 962 | 39.5% (380) | 39.5% (380) | 31.9% (307) |
| person occupation | 31 | 100.0% (31) | 100.0% (31) | 93.5% (29) |
| person university | 69 | 53.6% (37) | 53.6% (37) | 50.7% (35) |
| pokemon evolution | 44 | 90.9% (40) | 90.9% (40) | 81.8% (36) |
| president birth year | 15 | 60.0% (9) | 60.0% (9) | 6.7% (1)$^*$ |
| president election year | 18 | 77.8% (14) | 77.8% (14) | 11.1% (2)$^*$ |
| product by company | 30 | 100.0% (30) | 100.0% (30) | 86.7% (26) |
| star constellation name | 31 | 93.5% (29) | 93.5% (29) | 87.1% (27) |
| superhero archnemesis | 90 | 84.4% (76) | 84.4% (76) | 81.1% (73) |
| superhero person | 100 | 89.0% (89) | 89.0% (89) | 84.0% (84) |
| task done by tool | 51 | 98.0% (50) | 98.0% (50) | 90.2% (46) |
| Relation where all $o$ is uniquely identified by the first token ($\times 26$) | — | 100% | 100% | 100% |
| **Average** | — | 93.17% | 93.17% | 92.17% |

**Single object assumption**    In some relations, there may be more than one correct answer (e.g., fruits often take many different outside colors). Our dataset only catalogs one canonical related object in each case. This does not impact our results because for each subject we focus on the single object *that the LM predicts*, but future work could extend our evaluation scheme to measure how well LREs estimate the LM's *distribution* of candidate objects.

Table 10: Relation-wise LRE performance on GPT-J, and respective hyperparameters.

| relation, $r$ | $|range_r|$ | $\ell_r$ | $\beta$ | $\rho_r$ | Faithfulness | Causality |
|---|---|---|---|---|---|---|
| adjective antonym | 95 | 8 | | 243 | 0.69 ± 0.07 | 0.86 ± 0.04 |
| adjective comparative | 57 | 10 | | 121 | 0.98 ± 0.01 | 0.94 ± 0.04 |
| adjective superlative | 79 | 10 | | 143 | 0.93 ± 0.02 | 0.99 ± 0.01 |
| characteristic gender | 2 | 1 | | 74 | 0.77 ± 0.11 | 0.97 ± 0.04 |
| city in country | 21 | 2 | | 115 | 0.44 ± 0.10 | 0.89 ± 0.09 |
| company CEO | 287 | 6 | | 173 | 0.06 ± 0.03 | 0.31 ± 0.05 |
| company hq | 163 | 6 | | 126 | 0.21 ± 0.06 | 0.49 ± 0.04 |
| country capital city | 24 | 3 | | 68 | 0.88 ± 0.07 | 0.99 ± 0.02 |
| country currency | 23 | 3 | | 88 | 0.58 ± 0.08 | 0.98 ± 0.03 |
| country language | 14 | 1 | | 63 | 0.88 ± 0.09 | 0.99 ± 0.03 |
| country largest city | 24 | 10 | | 74 | 0.92 ± 0.05 | 0.99 ± 0.02 |
| food from country | 26 | 3 | | 113 | 0.51 ± 0.12 | 0.97 ± 0.05 |
| fruit inside color | 6 | 7 | | 107 | 0.65 ± 0.15 | 0.93 ± 0.07 |
| fruit outside color | 9 | 5 | | 160 | 0.78 ± 0.15 | 0.83 ± 0.12 |
| landmark in country | 91 | 6 | | 97 | 0.36 ± 0.06 | 0.68 ± 0.02 |
| landmark on continent | 5 | 4 | | 158 | 0.56 ± 0.13 | 0.91 ± 0.02 |
| name birthplace | 8 | 7 | | 91 | 0.92 ± 0.05 | 0.96 ± 0.07 |
| name gender | 2 | emb | | 17 | 0.80 ± 0.16 | 0.94 ± 0.04 |
| name religion | 5 | 4 | | 57 | 0.80 ± 0.10 | 0.99 ± 0.02 |
| object superclass | 10 | 7 | | 91 | 0.85 ± 0.05 | 0.93 ± 0.03 |
| occupation age | 2 | 5 | | 34 | 0.68 ± 0.03 | 1.00 ± 0.00 |
| occupation gender | 2 | 4 | | 34 | 0.98 ± 0.04 | 1.00 ± 0.00 |
| person father | 968 | 8 | 2.25 | 217 | 0.07 ± 0.03 | 0.28 ± 0.04 |
| person lead singer of band | 21 | 8 | | 163 | 0.64 ± 0.09 | 0.84 ± 0.09 |
| person mother | 962 | 6 | | 170 | 0.14 ± 0.04 | 0.39 ± 0.05 |
| person native language | 30 | 6 | | 92 | 0.65 ± 0.16 | 0.87 ± 0.03 |
| person occupation | 31 | 8 | | 131 | 0.49 ± 0.08 | 0.66 ± 0.06 |
| person plays instrument | 6 | 9 | | 198 | 0.59 ± 0.10 | 0.76 ± 0.04 |
| person sport position | 14 | 5 | | 97 | 0.42 ± 0.15 | 0.74 ± 0.05 |
| person university | 69 | 4 | | 153 | 0.64 ± 0.11 | 0.91 ± 0.04 |
| plays pro sport | 5 | 6 | | 117 | 0.76 ± 0.06 | 0.94 ± 0.01 |
| pokemon evolution | 44 | 7 | | 206 | 0.15 ± 0.05 | 0.25 ± 0.08 |
| president birth year | 15 | 6 | | 106 | 0.54 ± 0.14 | 0.84 ± 0.08 |
| president election year | 18 | emb | | 84 | 0.52 ± 0.20 | 0.91 ± 0.09 |
| product by company | 30 | 4 | | 158 | 0.31 ± 0.14 | 0.54 ± 0.05 |
| star constellation name | 31 | 8 | | 152 | 0.41 ± 0.08 | 0.27 ± 0.04 |
| substance phase of matter | 3 | 7 | | 60 | 0.87 ± 0.09 | 0.97 ± 0.03 |
| superhero archnemesis | 90 | 11 | | 192 | 0.30 ± 0.08 | 0.60 ± 0.10 |
| superhero person | 100 | 8 | | 228 | 0.44 ± 0.06 | 0.71 ± 0.07 |
| task done by tool | 51 | 5 | | 145 | 0.29 ± 0.10 | 0.76 ± 0.07 |
| task person type | 32 | 8 | | 109 | 0.49 ± 0.10 | 0.77 ± 0.10 |
| verb past tense | 76 | 11 | | 182 | 0.95 ± 0.03 | 0.97 ± 0.02 |
| word first letter | 25 | 7 | | 121 | 0.58 ± 0.09 | 0.91 ± 0.02 |
| word last letter | 18 | 6 | | 61 | 0.57 ± 0.13 | 0.83 ± 0.10 |
| word sentiment | 3 | 4 | | 94 | 0.63 ± 0.16 | 0.93 ± 0.03 |
| work location | 24 | 5 | | 112 | 0.55 ± 0.09 | 0.94 ± 0.06 |

