# OpenReview forum: "Linearity of Relation Decoding in Transformer Language Models"
_ICLR.cc/2024/Conference — ICLR 2024 spotlight_

### Official Review · Reviewer_zSUm · 2023-10-31

**Soundness:** 3 good
**Presentation:** 4 excellent
**Contribution:** 4 excellent
**Rating:** 8
**Confidence:** 3

**Summary:**

*update* The authors have addressed most weaknesses/errors that I have raised. I updated my score upwards as a result. */update*

It is well known that language models acquire knowledge of subjects, objects and their relations. However, to date, it is not well understood how these relations are represented in the model. The paper posits that relations are implemented as Linear Relational Embeddings, that is, affine transformations mapping from the embedding of the subject to the embedding of the object. The paper proposes to compute these mappings from the Jacobian $\delta o / \delta s$, where o and s are object and subject embeddings at certain layers, respectively.
The paper validates this approach by measuring a) faithfulness, i.e., whether decoding from the LRE mapping matches decoding from the true model when given a prompt expressing the relation, and b) to what extend this mapping is a causal explanation by providing an inverse of the LRE capable to change s such that a different, desired o results.
The experiments show that the method results in relatively high, but far from perfect, faithfulness and causality scores. The method is compared to number of reasonable baselines/ablations. Finally, the method is applied to the use case of detecting when the language model outputs false relationships despite knowing the true relationship.

**Strengths:**

* the paper covers an important topic that will certainly raise interest among the ICLR attendees
* the methods and results are interesting - I learned something.
* the presentation is excellent. The paper is easy to follow and results are presented in a comprehensive fashion.

**Weaknesses:**

* Limitations are not discussed in enough detail. For example, given a fixed subject and relation, it is possible that there are multiple true objects. This has consequences both for the generation of the datasets (where examples are filtered out if they're not generated by the model) and questions to what extend an invertible function can be a reasonable approximation of the true function, since invertibility implies that the function/relation is injective. Discussing these aspects would improve the paper.

I am a bit skeptical with regards to some results:
* Table 4 (Appendix A) shows the number of correctly predicted relations per language model. It is quite remarkable that some models in some categories get zero percent of the facts right, whereas other models get a much larger percentage right. Zero percent hints at systematic errors that remain undiscussed in the paper. For example, the LLaMA-13B achieves zero percent on the task of predicting president's birth year and election year. This is quite curious, given that these are quite well-known facts that appear a lot in the pretraining data, and the other models get them right. There is a reason for the poor performance of LLaMA on this particular task: Their tokenizer represents every digit as its own token. However, the caption of Table 4 states that only the first token was used to determine correctness. Since the LLaMa model would require 4 tokens, it had no chance of predicting the "correct" token. IMO, this is shortcoming of the proposed measurement, and accepting this bias should at least be discussed. I suspect that similar reasons might cause zero percent accuracy on some of the other tasks.
* Table 4 (Appendix A) shows that GPT-J gets zero examples from the "occupation gender" category right. Hence, none of these examples from this category should remain in the dataset for GPT-J according to the description in Section 4 (Dataset). Curiously, Figure 3 shows that "occupation gender" has a near 100% faithfulness success rate. How can that be if there are no such examples?

**Questions:**

Please respond to the weaknesses raised. I am happy to raise my score if the explanations are satisfactory.

---

> ### Author Response · Authors · 2023-11-18
>
> Thanks for the review and your many helpful comments! We are glad to hear that you found our work insightful and interesting, and we think we have addressed your concerns in our revision.
>
> ---
>
> > Limitations are not discussed in enough detail.
>
> Thanks for the suggestion. We have added a Limitations discussion in the appendix (Appendix I) that discusses dataset size, the risk for false positives when looking only at first tokens, and the assumption that every subject maps to a single object.
>
> ---
>
> > For example, given a fixed subject and relation, it is possible that there are multiple true objects. This has consequences both for the generation of the datasets (where examples are filtered out if they're not generated by the model) and questions to what extend an invertible function can be a reasonable approximation of the true function, since invertibility implies that the function/relation is injective. Discussing these aspects would improve the paper.
>
> You are correct that some of the relations we investigate can have one-to-many mapping where our dataset only records one object for each example. We have discussed this issue in Limitations (Appendix I).
>
> Our measurements are based on the following observation: the role of the LRE is not to recover the idealized relation, but **the LM’s predictions for that relation**. An LM presented with any specific sentence will assign the highest probability on **one** specific token, so the approach we have taken asks: can an LRE recover that same token? By its nature, the top-token prediction is well-defined (each subject maps to one object).
>
> ---
>
> > Table 4 (Appendix A) shows that GPT-J gets zero examples from the "occupation gender" category right.
>
> Good catch! It turns out there was a bug in the generation of Table 4: instead of collating our full experiment setup on $n = 8$ examples across 24 trials, we generated it based on $n = 5$ examples on a single trial, which erroneously resulted in some tabulated zeros. We have fixed the script and updated Table 4, and now there are no cells with zeros.
>
> ---
>
> > For example, the LLaMA-13B achieves zero percent on the task of predicting president's birth year and election year.
>
> We appreciate the thorough error analysis! Indeed, LLaMA’s tokenizer separates each digit in the years, Since we only look at the first token of both the true object and the predicted object to determine correctness, it is trivial in the opposite direction: the first token of the true object is always `"1"` for *president birth year*, so LLaMA need only predict `"1"` (which it does). This is also true for most of the examples in *president election year*. The reason LLaMA was scoring 0 in Figure 18 is because we weren’t accounting for spacing in front of the `"1"`.
>
> Note that for these cases specifically, **we had already excluded them from the faithfulness and causality results in the original draft**. We realize this was very unclear in the text, so we have added extra wording to Tables 4, 9 and Figure 18  to explain why LLaMA was not evaluated on relations where the object is a year. Additionally, in our new limitations section we discuss this as a pitfall of the first-token evaluation procedure.

---

> > ### Author Response · Authors · 2023-11-22
> >
> > Hi Reviewer zSUm,
> >
> > We have a short time left in the discussion period, so a gentle nudge to please let us know if we have addressed your questions or if you have any further questions. As a reminder, we have discussed several limitations of our work in Appendix I as per your suggestion. We have also fixed Table 4 (thank you for reporting the bug) and changed Table 18 to clarify certain issues with LLaMA tokenizer on relations *president birth year* and *election year*. We feel these updates have strengthened our paper. We invite you to examine these updates and consider increasing your score if your questions have been addressed.
> >
> > Thank you!

---

> > > ### Comment · Area_Chair_h3eq · 2023-12-03
> > >
> > > Hi reviewer zSUm. Can you kindly respond to the authors' rebuttal and discuss / let us know if you'd like to revise your score. Thanks.

---

> > ### Comment · Reviewer_zSUm · 2023-12-04
> >
> > Thank you, your clarifications have made the paper stronger. For my liking you should be more explicit about the limitations in the main text (adding 1-2 sentences to where you reference appendix I), but I'll recommend accepting this paper.
> >
> > Congratulations on your fine work!

---

### Official Review · Reviewer_y12e · 2023-11-01

**Soundness:** 3 good
**Presentation:** 4 excellent
**Contribution:** 3 good
**Rating:** 8
**Confidence:** 3

**Summary:**

This paper analyzes how the knowledge of relational triples (i.e., subject-relation-object) can be decoded in large language models (LLMs). Specifically, the authors hypothesize that the object embedding can be computed using an affine function (i.e. LRE) from the subject embedding and show that this holds for the majority of relations tested in this work. They also show the causality of the relationship through intervention experiments where a different object can be obtained by changing the subject embedding according to the LRE.  The authors also present the Attribute Lengs, an interesting application of their observation similar to the Logit Lens, which allows one to examine what entities are predicted to be the object in intermediate layers of a transformer model.

**Strengths:**

- The paper presents a novel insight into how relational knowledge is represented in an LLM, which should contribute to a deeper understanding of LLMs in the field.
- Extensive experiments are carried out to confirm the findings.
- The paper is well written and easy to follow

**Weaknesses:**

- The configuration for estimating the parameters of the affine functions could be further explored.

**Questions:**

- In experiments, n = 8 examples are used to estimate W and b. Would it be difficult to use more examples?  They seem to be too few to reliably estimate W and b.  I would also be interested in the variance as well as the mean.
- Does a “sample” mean a single example?  In statistics, a sample usually means a collection of examples (data points).
- p. 4: of LM’s decoding -> of the LM’s decoding?
- p. 5: by (Merullo et al., 2023) -> by Merullo et al. (2023)?
- p. 7: is a higher -> is higher?
- p. 8: visualizes a next -> visualizes next?

---

> ### Author Response · Authors · 2023-11-18
>
> Thank you for the positive review! We are pleased to know that you found our work novel and insightful.
>
> ---
>
> > In experiments, n = 8 examples are used to estimate W and b. Would it be difficult to use more examples? They seem to be too few to reliably estimate W and b. I would also be interested in the variance as well as the mean.
>
> We find that for most of the relations LRE performance saturates after $n = 5$. We have included a Figure (Figure 12) in the appendix which shows our measurements up to $n = 11$, and the plateau after 5. The reason that our LRE estimation procedure is so sample efficient is because it is estimated as a **first-order approximation of the LM computation itself**. We don’t *train* a linear classifier (or affine transformation) to match a set of input-output pairs. Our choice of $n=8$ for large-scale experiments is largely a practical one, determined by the memory consumption of GPT-J on the A6000 GPU available for our research.
>
> We agree the variance is also important to visualize: Table 10 (previously Table 8) in our appendix shows the standard deviation in LRE performance across 24 trials, each time calculated on different sets of training examples.
>
> ---
>
> > Does a “sample” mean a single example? In statistics, a sample usually means a collection of examples (data points).
>
> Yes! In our draft by sample we mean a single example or a datapoint. To avoid confusion, we have updated the paper to call these “examples”.
>
> ---
>
> > various typos
>
> Thanks for catching these typos.  We have addressed all of them in the revision.

---

> > ### Comment · Reviewer_y12e · 2023-11-22
> >
> > Thank you for the response and the revision. I remain positive about the paper and would like to keep my score.

---

### Official Review · Reviewer_wBgc · 2023-11-02

**Soundness:** 3 good
**Presentation:** 3 good
**Contribution:** 2 fair
**Rating:** 6
**Confidence:** 4

**Summary:**

This work focuses on analyzing the computation of LLM in the tasks of knowledge decoding. The authors find that a certain kind of computation in relation decoding can be approximated by linear relational embeddings such that R(s) = \beta Ws + b. Specifically, the intermediate hidden representation of subject is used in this linear transformation. With experiments on 47 different relations, the authors find that for some relations the linear approximation hold. However, this linear attribute is not universal. Furthermore, the authors conducted experiments to show the causality of the linearity of relation decoding. Finally, a visualization tool called attribute lens is proposed to show where and when the LM finishes retrieving knowledge about a specific relation.

**Strengths:**

This work focused on an interesting question about what computations the LMs perfrom while resolving relations. The authors smartly use the local derivative to obtain the affine transformation approximation. This is aligned with the traditional design in training knowledge graph embedding. Also, it is very useful to show that causality of the linearly decoding behavior, by using a low-rank pseudoinverse to obtain the perturbation of subject.

**Weaknesses:**

It is unknown that how the context prompt (e.g., [s] plays the) affects the conclusion. For example, will the conclusion hold if we change some other contexts that express same meaning of relation? Though it is interesting to show that some relations matches the linearity hypothesis (e.g., occupation gender, adjective comparative) which aligning traditional methods of training knowledge graph embedding, my concern is that the faithfulness is not very high for most of relations. This paper is more like a case study instead of a systematic measurement which covers a broader range of relations. It is hard to conclude how much percent of relations match the hypothesis.
Furthermore, it is unclear, if I did not miss some texts, why the causality is usually higher than faithfulness.

**Questions:**

In term of handling objects that have multiple tokens, I saw in Table 4: only the first token was used to determine correctness. Do we apply this strategy to all experiments? Will this lead to false positive? For causality experiment, do we also use the first token of o’ for experiment?
Do we have some examples showing that `when LRE can not fully capture the LM's computation of the relation, the linear approximation can still perform a successful edit`?
In which level of faithfulness, we can say that the linearity holds for the relation?

---

> ### Author Response · Authors · 2023-11-18
>
> Thank you for your constructive review. We are pleased to know that you found the question we are investigating to be “interesting” and our findings to be “useful”!
>
> ---
>
> > It is unknown that how the context prompt (e.g., [s] plays the) affects the conclusion. For example, will the conclusion hold if we change some other contexts that express same meaning of relation?
>
> During early versions of our experiments, we averaged results over multiple different prompts to account for potential difference in LRE performance. We later decided to use only one prompt per relation due to time constraints and because we noticed similar LRE performance across different prompts. We realize this confound is important to account for, so we’ve added Table 7 to the appendix which shows how faithfulness and causality scores vary when LRE is estimated with different prompt templates. Ultimately, we find that most scores do not change dramatically.
>
> ---
>
> > In term of handling objects that have multiple tokens, I saw in Table 4: only the first token was used to determine correctness. Do we apply this strategy to all experiments? Will this lead to false positives? For causality experiment, do we also use the first token of o’ for experiment?
>
> We *always* use the first token for the objects that get tokenized to multiple tokens. This is true for the causality experiments as well.  For some specific relations this strategy can indeed lead to false positives, though we find in practice that such collisions are rare.
>
> To clarify this, we have made two adjustments to our submission. First, we now discuss this issue explicitly in the limitations. Second, we have included a new table (Table 9) in the appendix which shows that first token collisions are rare: on average, over $93$% of the objects per relation in our dataset can be uniquely identified with their first token. It is worth noting that this number is skewed downwards by relations such as *person mother*, *person father*, and *president birth year*. And, the LRE performs poorly on these relations anyway.
>
> ---
>
> > Do we have some examples showing that `when LRE can not fully capture the LM's computation of the relation, the linear approximation can still perform a successful edit`?
>
> Here is one illustrative example where the LRE is not faithful yet we have observed a successful causal edit. Given the prompt `"Malcolm MacDonald works as a"`, GPT-J correctly predicts the expected answer, `" politician"`. However, an LRE (estimated based on the following examples) will assign the highest probability to `" writer"`.
>
> ```
> [
>     (subject='Janet Gunn', object='actor'),
>     (subject='Tony Pua', object='politician'),
>     (subject='Valentino Bucchi', object='composer'),
>     (subject='Christopher Guest', object='comedian'),
>     (subject='Giovanni Battista Guarini', object='poet'),
>     (subject='Menachem Mendel Schneerson', object='rabbi'),
>     (subject='Cristina Peri Rossi', object='novelist'),
>     (subject='Nils Strindberg', object='photographer')
> ]
> ```
> Although the correct answer `" politician"`  is within the top-5 prediction of the LRE, our faithfulness metric stringently measures top-1 accuracy and will consider it a failure case.
>
> In contrast, when we compute the edit direction as  $\Delta \mathbf{s} = W_r^{-1}(\mathbf{o}' - \mathbf{o})$, we find that $\mathbf{s} + \Delta \mathbf{s}$ is often enough to change the LM output from $o$ to $o'$ even when LRE($\mathbf{s'}$) does not assign highest probability to $o’$.
>
> Continuing this example: LRE failed to predict the correct occupation of $s’$ = `"Malcolm MacDonald"`. However, we can use GPT-Js object representation $\mathbf{o'}$ corresponding to its encoding of Malcom Macdonald’s profession to successfully change the profession of $s$ = `"Walter Hines Page"`, from $o$ = `" journalist"` to $o’$ = `" politician"`, which matches the correct profession of `"Malcom Macdonald"`, even though LRE($\mathbf{s’}$) incorrectly predicted `" writer"`.
>
> When causality exceeds faithfulness, it reflects relations for which such examples are prevalent.
>
> ---

---

> > ### Author Response · Authors · 2023-11-18
> >
> > > In which level of faithfulness, we can say that the linearity holds for the relation?
> >
> > Our LRE faithfulness measure is intended to be a fine-grained measure of linearity that is more informative than a binary assessment of “linear” vs “nonlinear”. In the paper, we say that relations for which the LRE attains at least 60% faithfulness are linearly decoded (meaning 22 out of 47 in our dataset are), but this is intended to help provide a summary of findings more than it is to suggest a hard cutoff. The answer is ultimately relation dependent, relying on the size of the range and how well the LM encodes that relation at all.
> >
> > That said, while it’s hard to provide an exact cutoff for when a relation is linearly decoded, we observe multiple cases where the relation is **unequivocally not** linearly decoded, including *person mother*, *person father*, *company CEO*, etc. LRE performance is close to random guessing on these relations, so we conclude that the model must employ some non-linear decoding mechanism instead.

---

> > > ### Author Response · Authors · 2023-11-22
> > >
> > > Hi Reviewer wBgc,
> > >
> > > We have a short time left in the discussion period, so a gentle nudge to please let us know if we have addressed your questions or if you have any further questions. As a reminder, we have modified Section 3.2, added Limitations (Appendix I) and Table 9 to discuss your concern with false positives while deciding accuracy with success on first-token prediction. We have also added Table 7 in Appendix F to show that LRE performs similarly when calculated with different prompt templates. We feel that these updates have strengthened our paper. We invite you to examine these updates and consider increasing your score if your questions have been addressed.
> > >
> > > Thank you!

---

### Author Response · Authors · 2023-11-18

We thank all the reviewers for their thoughtful and constructive feedback. We have updated our submission to incorporate this feedback, which we believe has substantially strengthened the submission. Here is a summary of our changes:
* [wBgc, zSUm] Added a Limitations section (Appendix I) detailing three core limitations of our approach mentioned in the reviews: dataset size, risk for false positives when looking only at first tokens, and the assumption that every subject maps to a single object.
* [wBgc, zSUm] Added explicit references to the first-token evaluation scheme and to the limitations section in Section 3.2.
* [wBgc] Added Table 7 showing how LRE performance does not vary dramatically when computed with different prompts.
* [y12e] Added Figure 12 showing that LRE performance saturates with ~5 or more training examples, to address the concern that more/fewer examples might change LRE performance.
* [wBgc] Added Table 9 showing number of unique objects per relation in our dataset, and also how many of them have unique first tokens when tokenized. This is to address the concern about deciding success based on first-token of the object leading to false positives, and we find that such collisions are rare for most relations.
* [zSUm] Fixed a bug in Table 4 that caused some models/relations to have a score of 0. Now all scores are nontrivial. Additionally, we identified two relations for LLaMA-13B where due to a weirdness with tokenization, the score will always be 1. We have ensured these relations are not included in our LLaMA-13B results and have updated the text to reflect it.
* [zSUm] Related to the bug above, we updated Figure 18 to gray out the removed relations.
* [y12e] Fixed some typos

We invite reviewers to reassess our submission as updated and to share any additional comments, questions, or suggestions. Please consider adjusting scores upwards if your concerns have been addressed.

We respond to each individual reviewer in separate responses.

---

### Meta-Review · Area_Chair_h3eq · 2023-12-10

**Metareview:**

The paper investigates the computation of relational knowledge in large language models (LLMs), proposing that relations can be approximated by linear transformations. The authors conduct experiments on 47 different relations, showing that linear approximations hold for some but not all relations. They explore the causality of the linear decoding behavior and introduce a visualization tool called "Attribute Lens" to examine the model's knowledge retrieval.

Strengths: The paper addresses an intriguing question about the computation of relations in LLMs, providing a novel insight into the representation of relational knowledge. The use of local derivatives for affine transformation approximation aligns with traditional knowledge graph embedding methods, enhancing the paper's methodological soundness. The experiments are extensive, and the paper is well-written and easy to follow, contributing to a deeper understanding of LLMs in the field.

Weaknesses:

1. The impact of context prompts on the conclusion's validity is unclear, because changes in context may affect the linearity hypothesis.
2. The paper is also challenged for being more of a case study than a systematic measurement, with doubts about the generalizability of results to a broader range of relations.
3. Reviewers also highlight potential biases in the evaluation metrics, noting cases where models achieve zero percent accuracy due to tokenization issues.

**Justification For Why Not Higher Score:**

While reviews were generally positive, the paper's measurement approach is questioned, and specific instances of systematic errors in certain tasks are pointed out, urging for a more thorough discussion of these limitations. Additionally, discrepancies between Table 4 and Figure 3 raise concerns about dataset integrity and the faithfulness metric, emphasizing the need for clarification.

**Justification For Why Not Lower Score:**

The experiments are extensive, and the paper is well-written and easy to follow, contributing to a deeper understanding of LLMs in the field.

---

### Decision · Program_Chairs · 2024-01-16

Accept (spotlight)